# Exponential concentration in quantum kernel methods

Supanut Thanasilp [1,2,3] ✉, Samson Wang [4], M. Cerezo [5,6] & Zoë Holmes[2,5] ✉

Kernel methods in Quantum Machine Learning (QML) have recently gained significant attention as a potential candidate for achieving a quantum advantage in data analysis. Among other attractive properties, when training a kernel-based model one is guaranteed to find the optimal model's parameters due to the convexity of the training landscape. However, this is based on the assumption that the quantum kernel can be efficiently obtained from quantum hardware. In this work we study the performance of quantum kernel models from the perspective of the resources needed to accurately estimate kernel values. We show that, under certain conditions, values of quantum kernels over different input data can be exponentially concentrated (in the number of qubits) towards some fixed value. Thus on training with a polynomial number of measurements, one ends up with a trivial model where the predictions on unseen inputs are independent of the input data. We identify four sources that can lead to concentration including expressivity of data embedding, global measurements, entanglement and noise. For each source, an associated concentration bound of quantum kernels is analytically derived. Lastly, we show that when dealing with classical data, training a parametrized data embedding with a kernel alignment method is also susceptible to exponential concentration. Our results are verified through numerical simulations for several QML tasks. Altogether, we provide guidelines indicating that certain features should be avoided to ensure the efficient evaluation of quantum kernels and so the performance of quantum kernel methods.

Quantum machine learning (QML) has generated tremendous amounts of excitement, but it is important not to over-hype its potential. On the one hand, a family of impressive results have recently established a provable separation between the power of classical and quantum machine learning methods in a range of contexts[1–10]. On the other, many proposals remain heuristic and there are significant questions yet to be answered on the efficient scalability of QML methods.

Quantum kernel methods, which involve embedding classical data into quantum states and then computing their inner-products

(i.e., their kernels), or in the case of quantum data directly computing input state overlaps, are widely viewed as particularly promising family of QML algorithms to achieve a practical quantum advantage. To ensure provable quantum speed-up over classical algorithms, the key is to construct the embedding (also called a quantum feature map) that is capable of recognizing classically intractable complex patterns[6–8]. Quantum kernels are expected to find use in a mix of scientific and practical applications including classifying types of supernovae in cosmology[11], probing phase transitions in quantum many-body

[1]Centre for Quantum Technologies, National University of Singapore, 3 Science Drive 2, Singapore. [2]Institute of Physics, Ecole Polytechnique Fédérale de Lausanne (EPFL), Lausanne, Switzerland. [3]Chula Intelligent and Complex Systems, Department of Physics, Faculty of Science, Chulalongkorn University, Bangkok, Thailand. [4]Imperial College London, London, UK. [5]Information Sciences, Los Alamos National Laboratory, Los Alamos, NM, USA. [6]Quantum Science Center, Oak Ridge, TN, USA. ✉e-mail: supanut.thanasilp@gmail.com; zoe.holmes@epfl.ch

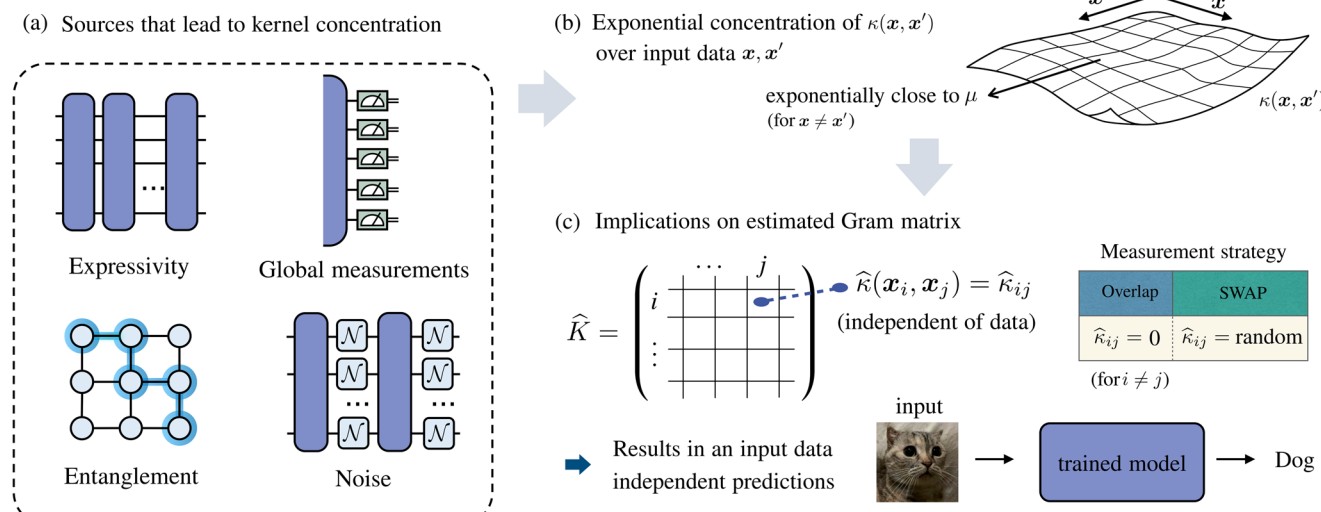

**Fig. 1 | Exponential concentration and its implications on kernel methods.** The exponential concentration (in the number of qubits $n$) of quantum kernels $\kappa(\boldsymbol{x},\boldsymbol{x}')$, over all possible input data pairs $\boldsymbol{x},\boldsymbol{x}'$, can be seen to stem from the difficulty of information extraction from data quantum states due to various sources (illustrated in panels **a** and **b**). The kernel concentration has a detrimental impact on the performance of quantum kernel-based methods. As shown in panel **c**, for a polynomial (in $n$) number of measurement shots, the statistical estimates of the off-diagonal elements in the Gram matrix $\hat{\kappa}(\boldsymbol{x}_i,\boldsymbol{x}_j)$ contain no information about the input data (with high probability) i.e., each $\hat{\kappa}(\boldsymbol{x}_i,\boldsymbol{x}_j) = \hat{\kappa}_{ij}$. The exact behavior of the estimated kernel value depends on the measurement strategy: for the Loschmidt Echo test (i.e., the overlap test), $\hat{\kappa}_{ij}$ concentrates to 0 for $i \neq j$ (corresponding to the estimated Gram matrix being an identity $\mathbb{1}$) and for the SWAP test $\hat{\kappa}_{ij}$ for $i \neq j$ is indistinguishable from a data-independent random variable (corresponding to the estimated Gram matrix being a random matrix). Ultimately, this leads to a trivial model where the predictions on unseen inputs are independent of the training data.

physics[12], and detecting fraud in finance[13]. Moreover, kernel methods are famously said to enjoy trainability guarantees due to the convexity of their loss landscapes[14–17].

This is in contrast to Quantum Neural Networks (QNNs) where the loss landscape is generally non-convex[18,19] and can exhibit Barren Plateaus (BPs). A barren plateau is a cost landscape where the magnitudes of gradients vanish exponentially with growing problem size[20–32]. There are a number of causes that can lead to barren plateaus, including using variational ansatze that are too expressive[20,26,33] or too entangling[23,34]. However, barren plateaus can even arise for inexpressive and low-entangling QNNs if the cost function relies on measuring global properties of the system[21] or if the training dataset is too random[29,35]. Hardware errors can also wash out landscape features leading to noise-induced barren plateaus[28,36].

Here we argue that quantum kernel methods experience a similar barrier to barren plateaus. Crucially, the trainability guarantees enjoyed by kernel methods only become meaningful when the values of the kernel can be efficiently estimated to a sufficient precision such that the statistical estimates contain information about the input data. We show that under certain conditions, the value of quantum kernels can exponentially concentrate (with increasing number of qubits) around a fixed value. In such cases, the number of shots required to resolve the kernels to a sufficiently high accuracy scales exponentially. This indicates that the efficient evaluation of quantum kernels cannot always be taken for granted. Consequently, when the kernel values are estimated with a polynomial number of measurements, the trained model with high probability becomes independent of input data. That is, the predictions of the model on unseen data are the same for any target problem that suffers from exponential concentration and thus the learnt model is, for all intents and purposes, useless. This is summarized in Fig. 1.

This concentration of quantum kernels can in broad terms be viewed as a result of the fact that it can be extremely difficult to extract any useful information from the (necessarily) exponentially large Hilbert space (especially in the presence of noise). We show that analogous to the causes of BPs for QNNs there are at least three different

mechanisms that can lead to the exponential concentration of the encoded quantum states, including (i) the expressivity of the encoded quantum state ensemble, (ii) the entanglement in encoded quantum states with a local observable and (iii) the effect of noise. We further show that for the case of the commonly used fidelity kernel[15,37], the dependence of global measurements to evaluate the kernel can lead to exponential concentration even when the expressivity of the embedding and the entanglement of the data states are low. In all cases, we establish exponential concentration by deriving an analytic bound (summarized in Table 1). We further provide numerical results demonstrating these effects for different learning tasks.

Our work on embedding-induced concentration suggests that problem-inspired embeddings should be used over problem-agnostic embeddings (which are typically highly expressive and entangling). For instance, one can construct embeddings encoding the geometrical properties of the data[38–42]. However, additional care should be taken if such embeddings are to be found through optimizing embedding architectures, since we show this training embedding process can also exhibit barren plateaus. Furthermore, we consider the projected quantum kernel which is constructed by measuring local subsystems and has been shown to maintain good generalization in a situation where the fidelity kernel fails to generalize[6,7].

In contrast to QNNs where the trainability barrier caused by BPs is now common knowledge, the community is generally less aware of the problems posed by exponential concentration for quantum kernel methods. The problem of exponential concentration for the fidelity quantum kernel was first observed in ref. 6 and later analyzed in refs. 7,43,44 in the context of generalization. ref. 7 discusses exponential concentration in the context of a projected quantum kernel for a specific example embedding. On the other hand, refs. 8,16 provide a rigorous study of the number of measurement shots required to successfully train the fidelity kernel but do not address the issue of exponential concentration. Here we provide a systematic treatment of the causes and effects of exponential concentration in the presence of shot noise. We intend our results to be viewed as a guideline to the types of kernels and embeddings to be avoided for successful training.

**Table 1 | Summary of our main results**

| Sources of concentration | Kernels | QNNs |
|---|---|---|
| Expressivity | Theorem 1 | refs. 20,25 |
| Global measurements | Proposition 3 | ref. 21 |
| Entanglement | Corollary 2 | refs. 23,34 |
| Noise | Theorem 3 | ref. 28 |

This table summarizes our key analytical results on different sources that lead to the exponential concentration in quantum kernels as compared with BP results of QNNs in the literature.

Moreover, our results on noise-induced kernel concentration serve as a warning against using deep encoding schemes in the near-term. For a more detailed survey of how our results fit in the context of prior work see Supplementary Note I.

## Results

### Framework

Our results apply generally to any method that involves quantum kernels. This includes both supervised learning tasks such as regression and classification tasks, as well as unsupervised learning tasks such as generative modeling and dimensional reduction. However, for concreteness we focus on supervised learning on classical data. Here, one is given repeated access to a training dataset $S := \{x_i, y_i\}_{i=1}^{N_s}$, where $x_i \in \mathcal{X}$ are input vectors and $y_i \in \mathcal{Y}$ are associated labels. The input vectors and labels are related by some unknown target function $f : \mathcal{X} \rightarrow \mathcal{Y}$. Our task is to use the dataset to train a parameterized QML model $h_a$, i.e. a function $h_a : \mathcal{X} \rightarrow \mathcal{Y}$ parameterized by $a$, to approximate $f$.

The model can be trained by introducing an empirical loss $\mathcal{L}_a$ which quantifies the degree to which the model $h_a$ agrees with the target function $f$ over the training data $S$. The optimal parameters of the model are given by

$$a_{\text{opt}} := \arg\min_a \mathcal{L}_a(S), \qquad (1)$$

and can be obtained by minimizing the empirical loss. Once trained, the model is tested on some unseen data. The hope is that if the dataset is sufficiently large and appropriately chosen, the optimized function $h_{a_{\text{opt}}}$ not only agrees on the training set but also accurately predicts the correct labels on unseen inputs. This is exactly the question of generalization[6–10,43–63]: does successful training on the training data imply good predictive power on unseen data?

In what follows we focus on quantum kernel methods. Here, each individual input data point $x_i$ is encoded into an $n$-qubit data-encoded quantum state $\rho(x_i)$ using a data-embedding unitary $U(x_i)$, so that

$$\rho(x_i) = U(x_i)\rho_0 U^\dagger(x_i), \qquad (2)$$

for some initial state $\rho_0$. Consequently, the training input dataset can be seen as an ensemble of data-encoded quantum states. For now, we leave the choice of $U(x_i)$ entirely arbitrary, and thus this framework includes all unitary embedding schemes.

For a given input data pair $x$ and $x'$ we evaluate a similarity measure $\kappa(x, x')$ between two encoded quantum states on a quantum computer. Formally, this is a function $\kappa : \mathcal{X} \times \mathcal{X} \rightarrow \mathbb{R}$ corresponding to an inner product of data states, and is known as a quantum kernel[6,15,37]. Here, we consider two common choices of quantum kernels. First, we study the fidelity quantum kernel[15,37], which is defined as

$$\kappa^{FQ}(x, x') = \text{Tr}[\rho(x)\rho(x')]. \qquad (3)$$

Second, we consider the projected quantum kernel[6], given by

$$\kappa^{PQ}(x, x') = \exp\left(-\gamma \sum_{k=1}^{n} \|\rho_k(x) - \rho_k(x')\|_2^2\right), \qquad (4)$$

where $\rho_k(x)$ is the reduced state of $\rho(x)$ on the $k$-th qubit, $\|\cdot\|_2$ is the Schatten 2-norm and $\gamma$ is a positive hyperparameter.

The power of kernel-based learning methods stems from the fact that they map data from $\mathcal{X}$ to a higher-dimensional feature space (in this case the $2^n$-dimensional Hilbert space) where inner products are taken and a decision boundary such as a support vector machine can be trained[64]. Notably, thanks to the Representer Theorem, the optimal kernel-based model is guaranteed to be expressed as a linear combination of the kernels evaluated over the training dataset (see Chapter 5 in ref. 64). More concretely, for a kernel-based QML model $h_a$ depends on the input data through the inner product between states. We have that the optimal solution is given by

$$h_{a_{\text{opt}}}(x) = \sum_{i=1}^{N_s} a_{\text{opt}}^{(i)} \kappa(x, x_i), \qquad (5)$$

where $a_{\text{opt}} = (a_{\text{opt}}^{(1)}, \ldots, a_{\text{opt}}^{(N_s)})$. Additionally, if the loss $\mathcal{L}$ is appropriately chosen, then the loss landscape can be guaranteed to be convex. It follows that by constructing the Gram matrix $K$ whose entries are kernels over training input pairs,

$$[K]_{ij} = \kappa(x_i, x_j), \qquad (6)$$

where $x_i, x_j \in S$, the optimal parameters $a_{\text{opt}}$ can be found by solving the convex optimization problem in Eq. (1). Thus if the Gram matrix can be calculated exactly, kernel-based methods are perfectly trainable.

As an example, in kernel ridge regression, we consider a square loss function $\mathcal{L}_a(S) = \frac{1}{2}\sum_{i=1}^{N_s}(h_a(x_i) - y_i)^2 + \frac{\lambda}{2}\|a\|_\mathcal{H}^2$ with a regularization $\lambda$ and a norm in a feature space $\|a\|_\mathcal{H}^2$. The optimal parameters can be analytically shown to be of the form

$$a_{\text{opt}} = (K - \lambda\mathbb{1})^{-1}y, \qquad (7)$$

where $y$ is a training label vector with the $i^{\text{th}}$ component $y_i$. Another common example is a support vector machine where we consider a binary classification problem with the corresponding labels $y \in \{-1, +1\}$. Using a hinge loss function with no regularization, i.e. $\mathcal{L}_a(S) = \frac{1}{N_s}\sum_{i=1}^{N_s}\max(0, 1 - h_a(x_i)y_i)$, the optimization problem in Eq. (1) can be reformulated as

$$a_{\text{opt}} = \arg\max_a\left[\sum_{i=1}^{N_s} a^{(i)} - \frac{1}{2}\sum_{i,j=1}^{N_s} a^{(i)}a^{(j)}y_iy_jK_{ij}\right], \qquad (8)$$

subject to $0 \leqslant a^{(i)}$ for all $i$. Assuming that the Gram matrix $K$ can be accurately and efficiently obtained, solving for the optimal parameters can done with a number of iterations in $\mathcal{O}(\text{poly}(N_s))$.

### Why exponential concentration is problematic

By virtue of their convex optimization landscapes, kernel methods are guaranteed to obtain the optimal model from a given Gram matrix. However, due to the probabilistic nature of quantum devices, in practice the entries of the Gram matrix can only be estimated via repeated measurements on a quantum device. Thus the model is only ever trained on a statistical estimate of the Gram matrix, $\hat{K}$, instead of the exact one, $K$. The resulting statistical uncertainty, as we will argue here, inhibits how well quantum kernel methods may perform.

The heart of the problem is that, in a wide range of circumstances, the value of quantum kernels exponentially concentrate. That is, as the size of the problem increases, the difference between kernel values become increasingly small and so, more shots are required to distinguish between kernel entries. With a polynomial shot budget this leads to an optimized model which is insensitive to the input data and cannot generalize well.

More generally, exponential concentration can be formally defined as follows.

**Definition 1.** (Exponential concentration) Consider a quantity $X(\boldsymbol{\alpha})$ that depends on a set of variables $\boldsymbol{\alpha}$ and can be measured from a quantum computer as the expectation of some observable. $X(\boldsymbol{\alpha})$ is said to be deterministically exponentially concentrated in the number of qubits $n$ towards a certain $\boldsymbol{\alpha}$-independent value $\mu$ if

$$|X(\boldsymbol{\alpha}) - \mu| \leqslant \beta \in O(1/b^n), \tag{9}$$

for some $b > 1$ and all $\boldsymbol{\alpha}$. Analogously, $X(\boldsymbol{\alpha})$ is probabilistically exponentially concentrated if

$$\Pr_{\boldsymbol{\alpha}}[|X(\boldsymbol{\alpha}) - \mu| \geqslant \delta] \leqslant \frac{\beta}{\delta^2}, \beta \in O(1/b^n), \tag{10}$$

for $b > 1$. That is, the probability that $X(\boldsymbol{\alpha})$ deviates from $\mu$ by a small amount $\delta$ is exponentially small for all $\boldsymbol{\alpha}$.

If $\mu$ additionally exponentially vanishes in the number of qubits i.e., $\mu \in \mathcal{O}(1/b'^n)$ for some $b' > 1$, we say that $X(\boldsymbol{\alpha})$ exponentially concentrates towards an exponentially small value.

We remark that using Chebyshev's inequality, probabilistic exponential concentration can also be diagnosed by analyzing the variance of $X(\boldsymbol{\alpha})$. That is, $X(\boldsymbol{\alpha})$ is exponentially concentrated towards its mean $\mu = \mathbb{E}_{\boldsymbol{\alpha}}[X(\boldsymbol{\alpha})]$ if

$$\mathrm{Var}_{\boldsymbol{\alpha}}[X(\boldsymbol{\alpha})] \in \mathcal{O}(1/b^n), \tag{11}$$

for $b > 1$, thus satisfying Definition 1. Here the variance is taken over $\boldsymbol{\alpha}$. If $0 \leqslant X(\boldsymbol{\alpha}) \leqslant 1$ for all $\boldsymbol{\alpha}$ (as for quantum kernels) one can demonstrate exponential concentration by showing that the mean $\mu = \mathbb{E}_{\boldsymbol{\alpha}}[X(\boldsymbol{\alpha})]$ is exponentially small which directly implies that $\mathrm{Var}_{\boldsymbol{\alpha}}[X(\boldsymbol{\alpha})] \in \mathcal{O}(1/b^n)$. Furthermore, in this context when $\mu$ vanishes exponentially, we can say that the probability of deviating from zero by an arbitrary constant amount is exponentially small.

Definition 1 is rather general and applies to a number of QML frameworks. In the case of quantum neural networks, $X(\boldsymbol{\alpha}) = C(\boldsymbol{\theta})$, where $\boldsymbol{\alpha} = \boldsymbol{\theta}$ and $C(\boldsymbol{\theta})$ is a cost function that depends on some variational ansatz parameters $\boldsymbol{\theta}$. In the context of quantum landscape theory, such concentration is central to studying the BP phenomenon. In particular, the equivalence between exponentially concentrating costs and vanishing gradients cost gradients is demonstrated in ref. 65. In the context of quantum kernels, the quantity of interest is the quantum kernel, i.e., $X(\boldsymbol{\alpha}) = \kappa(\boldsymbol{x}, \boldsymbol{x}')$ where the set of variables is a pair of input data $\boldsymbol{\alpha} = \{\boldsymbol{x}, \boldsymbol{x}'\}$. Hence the probability in Eq. (10) and the variance in Eq. (11) is now taken over all possible pairs of input data $\{\boldsymbol{x}, \boldsymbol{x}'\}$.

To understand the problems caused by exponential concentration, let us first consider the fidelity kernel. In practice, the kernel value is statistically estimated from measuring $N$ samples where (on all but classically simulable quantum devices) we assume we are restricted to $N \in \mathcal{O}(\mathrm{poly}(n))$. For a given input data pair $\boldsymbol{x}$ and $\boldsymbol{x}'$, we consider two common measurement strategies to estimate the kernel value: (i) the Loschmidt Echo test (i.e., the overlap test) and (ii) the SWAP test. In either case, the fidelity quantum kernel is equivalent to the expectation value of an observable $O$ for some quantum state $\rho$ with the exact expression for $O$ and $\rho$ depending on the strategy used. If we write the eigendecomposition of the observable as $O = \sum_i o_i |o_i\rangle \langle o_i|$ where $o_i$ and $|o_i\rangle$ are the eigenvalues and eigenvectors of $O$ respectively, then the statistical estimate after $N$ measurements is given by

$$\widehat{\kappa}^{\mathrm{FQ}}(\boldsymbol{x}, \boldsymbol{x}') = \frac{1}{N} \sum_{m=1}^{N} \lambda_m. \tag{12}$$

Here $\lambda_m$ is the outcome of the $m^{\mathrm{th}}$ measurement and can be treated as a random variable which takes the value $o_i$ with probability $p_i = \mathrm{Tr}[|o_i\rangle \langle o_i|\rho]$.

The behavior of the statistical estimate depends on the measurement strategy taken. When employing the Loschmidt Echo test, the kernel value corresponds to the probability of observing the all-zero bitstring. To estimate this probability, we assign $+1$ to the outcome of obtaining the all-zero bitstring and assign 0 to other bitstrings. If the kernel value concentrates to an exponentially small value i.e., $\mu \in \mathcal{O}(1/b^n)$, then the chance of never obtaining the all-zero bitstring from $N$ samples is $(1-\mu)^N \approx 1 - N\mu$. That is, with a polynomial number of samples $N \in \mathcal{O}(\mathrm{poly}(n))$, it is very likely that none are the all-zero bitstring and hence likely that the statistical estimate of the kernel is zero. This is formalized in the following proposition (proven in Supplementary Note III A 1).

**Proposition 1.** Consider the fidelity quantum kernel as defined in Eq. (3). Assume that the kernel values $\kappa^{\mathrm{FQ}}(\boldsymbol{x}, \boldsymbol{x}')$ exponentially concentrate towards an exponentially small value as per Definition 1. Supposing an $N \in \mathcal{O}(\mathrm{poly}(n))$ shot Loschmidt Echo test is used to estimate the Gram matrix for a training dataset $\mathcal{S} = \{\boldsymbol{x}_i, y_i\}$ of size $N_s$ then, with a probability exponentially close to 1, the statistical estimate of the Gram matrix $\widehat{K}$ is equal to the identity matrix. That is,

$$\Pr\left[\widehat{K} = \mathbb{1}\right] \geqslant 1 - \delta', \delta' \in \mathcal{O}(c^{-n}) \tag{13}$$

for some $c > 1$.

In the case of the SWAP test the measurement outcomes are either $+1$ with probability $p_+ = 1/2 + \kappa^{\mathrm{FQ}}(\boldsymbol{x}, \boldsymbol{x}')/2$, or $-1$ with probability $1 - p_+$. Thus computing the kernel value amounts to determining the perturbation from the uniform distribution where the $+1$ and $-1$ outcomes occur with equal probabilities. Intuitively, when the kernel value concentrates to an exponentially small value, the perturbation cannot be detected with a polynomial number of measurement shots. In other words, a statistical estimate using only a polynomial number of shots does not contain information about the input data pair with probability exponentially close to 1. This is formally stated in the following proposition which is derived by reducing the problem of distinguishing distributions (i.e., one associated with a kernel value and the uniform distribution) to a hypothesis testing task.

**Proposition 2.** Assume that the fidelity quantum kernel $\kappa^{\mathrm{FQ}}(\boldsymbol{x}, \boldsymbol{x}')$ exponentially concentrates towards some exponentially small value as per Definition 1. Suppose an $N \in \mathcal{O}(\mathrm{poly}(n))$ shot SWAP test is used to estimate the Gram matrix for a training dataset $\mathcal{S} = \{\boldsymbol{x}_i, y_i\}$ of size $N_s$. Then, with probability exponentially close to 1 (i.e., probability at least $1 - \delta'$ such that $\delta' \in \mathcal{O}(c^{-n})$ for some $c > 1$), the estimate of the Gram matrix $\widehat{K}$ is statistically indistinguishable from the matrix $\widehat{K}_N^{(\mathrm{rand})}$ whose diagonal elements are 1 and off-diagonal elements are instances of

$$\widehat{\kappa}_N^{(\mathrm{rand})} = \frac{1}{N} \sum_{m=1}^{N} \widetilde{\lambda}_m, \tag{14}$$

where each $\widetilde{\lambda}_m$ takes either $+1$ or $-1$ with equal probability. We note that $\widehat{\kappa}_N^{(\mathrm{rand})}$ does not contain any information about the input data $\mathcal{S} = \{\boldsymbol{x}_i, y_i\}$.

We refer the readers to Supplementary Note II for an introduction to some preliminary tools for a hypothesis testing and Supplementary Note III A 2 for further technical details regarding the SWAP test, which includes formal definitions of statistical indistinguishability (i.e., Definition 2 for distributions and Definition 3 for outputs), and a proof of the proposition.

Although statistical estimates of the kernel behave differently depending on the choice of measurement strategy, they are both in effect independent of the input data for large $n$. Thus training with this estimated Gram matrix leads to a model whose predictions are independent of the input training data. We present numerical simulations to support our theoretical findings in Supplementary Note III A 3.

Crucially, this conclusion applies generally beyond kernel ridge regression to other kernel methods including both supervised learning tasks and unsupervised learning tasks. As a concrete example, we consider the optimal solution for kernel ridge regression in the presence of exponential concentration.

**Corollary 1.** Consider a kernel ridge regression task with a squared loss function and regularization $\lambda$ using the same assumptions as Proposition 1. Denote $\mathbf{y}$ as a vector with its $i^{\text{th}}$ elements equal to $y_i$.

1. For the Loschmidt Echo test, the optimal parameters are found to be

$$\mathbf{a}_0(\mathbf{y}, \lambda) = \frac{\mathbf{y}}{1-\lambda}, \tag{15}$$

with probability at least $1 - \delta$ with $\delta \in \mathcal{O}(b^{-n})$ for some $b > 1$.

For a test data point $\mathbf{x} \notin \mathcal{S}$, the model prediction is 0 with probability at least $1 - \delta'$ such that $\delta' \in \mathcal{O}(b'^{-n})$ for some $b' > 1$.

2. For the SWAP test, the optimal parameters are statistically indistinguishable from the vector

$$\mathbf{a}_{\text{rand}}(\mathbf{y}, \lambda) = \left(\widehat{K}_N^{(\text{rand})} - \lambda \mathbb{1}\right)^{-1} \mathbf{y}, \tag{16}$$

with probability at least $1 - \tilde{\delta}$ with $\tilde{\delta} \in \mathcal{O}(\tilde{b}^{-n})$ for some $\tilde{b} > 1$. Here, $\widehat{K}_N^{(\text{rand})}$ is a data-independent random matrix whose diagonal elements are 1 and off-diagonal elements are instances of $\widehat{\kappa}_N^{(\text{rand})}$ in Eq. (14).

In addition, with probability exponentially close to 1, the model prediction on unseen data is statistically indistinguishable from the data-independent random variables that result from measuring $\widehat{K}_N^{(\text{rand})}$.

Corollary 1 shows that, regardless of the measurement strategy to estimate the kernel value, exponential concentration leads to a trained model where the predictions on unseen inputs are independent of the training data. A visual illustration of the effect of exponential concentration in the presence of shot noise on model predictions is provided in Fig. 2. We note that these propositions and corollary presented here are simplified versions and refer the readers to Supplementary Note III A for the full statements and proofs.

It is natural to ask whether the problems caused by exponential concentration should be viewed as a barrier to training or generalization. Since the estimated Gram matrix is still positive semi-definite, the loss landscape remains convex when the model is trained and the optimal model with respect to this estimated Gram matrix is guaranteed to be obtained. Although this trained model is independent of input data (as explained above), the model can still perform well on the training phase and achieve small training errors in the limit of small regularization. This is because the training output data is trivially 'cooked' into the model via the optimization process (independently of the kernel values).

On the other hand, the data independence of the kernel values means that the predictions of the trained model are completely independent of the training data and so the trained model in general performs trivially on unseen data. That is, the model generalizes terribly. By incorporating the effect of shot noise, this has a different flavor to typical barriers to generalization in that crucially it arises from using not enough shots (rather than not enough training data). Moreover, in

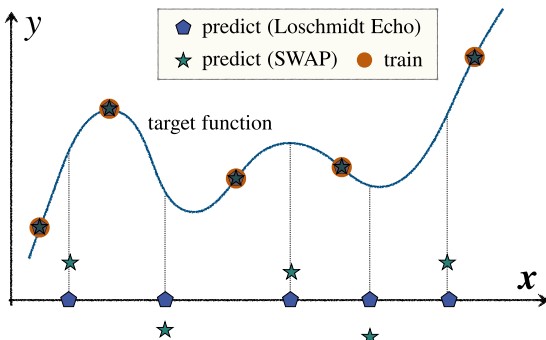

**Fig. 2 | Schematic of effect of exponential concentration and shot noise on training and generalization performance.** For the unseen (test) data, the behavior depends on how kernel values are statistically estimated. In the case of the Loschmidt Echo test, the model predictions are zero with high probability. On using the SWAP test, the model predictions fluctuate around zero (due to shot noise). On the other hand, for the training data, the training labels are effectively hard-coded by the optimization process. (For simplicity we here consider the limit of no regularization).

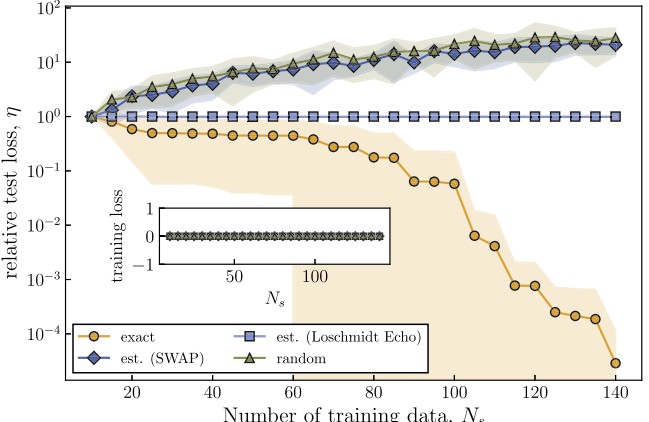

**Fig. 3 | Effect of exponential concentration on training and generalization performance.** We consider a tensor product encoding for an engineered data set where each component is uniformly drawn from $[0, 2\pi]$ and the true label is $y_{\text{true}}(\mathbf{x}) = \sum_{i=1}^{N_s} w_i \kappa^{\text{FQ}}(\mathbf{x}_i, \mathbf{x})$ where $w_i$ is randomly chosen from $[0, 1]$. We train on $N_s = 150$ data points. In the main plot, the loss on a test dataset $\mathcal{S}_{\text{test}}$ relative to its initial value (without training) is plotted as a function of increasing training data. In the inset, an absolute training error is plotted as a function of the increasing data. We note that each kernel value is estimated with $N = 1000$ and the number testing data points is 20. The training is done with no regularization $\lambda = 0$. We repeat this experiment 10 times. The solid curves represent averages of respective losses and the shaded areas represent standard deviations.

our numerics below we concretely see that this barrier cannot be resolved by training on more input data points.

Figure 3 numerically demonstrates this effect on an engineered dataset for a 40 qubit simulation. In the main plot, the generalization is studied as a function of increasing training data $\mathcal{S}_{N_s}$ and whether the training is performed with exact or estimated kernel values. Particularly, to observe the improvement due to increasing data, we plot a relative loss on a test dataset $\mathcal{S}_{\text{test}}$ with respect to its initial value $(N_S = 10)$ i.e., $\eta(N_s) = \frac{\mathcal{L}_{\mathbf{a}}(\mathcal{S}_{\text{test}} | \mathcal{S}_{N_s})}{\mathcal{L}_{\mathbf{a}}(\mathcal{S}_{\text{test}} | \mathcal{S}_{N_s = 10})}$. That is, $\eta(N_s) < 1$ for $N_s > 10$ indicates better generalization with increasing training data. This is observed to be the case for the training on the exact kernel value where the model gradually generalizes better. In fact, this learning task is synthesized such that when training on the whole dataset, with an access to the exact kernel values, the trained model generalizes perfectly. Even with this dataset which is heavily favorable for the fidelity kernel, the

performance on unseen data with the estimated kernel values shows no improvement with the increasing training data. Specifically, when the Loschmidt Echo test is used to evaluate kernel values, the statistical estimates accumulate at exactly zero, leading to $\eta(N_s) = 1$ for all $N_s$. In addition, for the SWAP measurement strategy, there is no improvement with increasing data and the behavior of $\eta(N_s)$ aligns with the one where the model is trained on a random matrix where each off-diagonal element is a data-independent random variable in Eq. (14). On the other hand, as demonstrated in the inset of Fig. 3, the trained model performs perfectly on the training set $\mathcal{S}_{N_s}$ and achieves zero training errors in all cases. This is again because the training label information is hard-coded in the optimization process. These empirical observations are all in good agreement with our theoretical predictions.

Finally, the analysis in the case of the projected quantum kernel is slightly more complicated as estimating the kernel requires us to first obtain the statistical estimates of the 2-norms between the reduced data encoding states on all individual qubits from quantum computers. Two common strategies to to do so include (i) the full tomography of the single qubit reduced density matrices and (ii) the local SWAP tests. In Supplementary Note III B, we again use a hypothesis testing framework to analyze the effect of exponential concentration on the projected kernel for these strategies. Similarly to the fidelity kernel we find that the final trained model is in effect independent of the training data.

## Sources of exponential concentration
Given that exponential concentration leads to trivial data-independent models, it is important to determine when kernel values will, or will not, concentrate. In this section, we investigate the causes of exponential concentration for quantum kernels.

In broad terms, the exponential concentration of quantum kernels may be viewed as stemming from the fact in certain situations it can be difficult to extract information from quantum states. In particular, we identify four key features that can severely hinder the information extraction process via kernels. These include the expressivity of the data embedding, entanglement, global measurements and noise (see Fig. 1). For each source, we derive an associated concentration bound. As summarized in Table 1 each of these theorems has an analog for QNN. All the proofs of our main results are presented in the Supplementary Information.

**1. Expressivity-induced concentration.** In broad terms, the expressivity of an ensemble of unitaries is defined as how close the ensemble uniformly covers the unitary group. To introduce the concept of the expressivity of the data embedding $U(\boldsymbol{x})$, we first consider the unitary ensemble generated via the data embedding $U(\boldsymbol{x})$ over all possible input data vectors $\boldsymbol{x} \in \mathcal{X}$. That is, the data embedding defines a map $U : \mathcal{X} \rightarrow \mathbb{U}_{\boldsymbol{x}} \subset \mathcal{U}(d)$, where $\mathcal{U}(d)$ is the total space of unitaries of dimension $d = 2^n$, and

$$\mathbb{U}_{\boldsymbol{x}} = \{U(\boldsymbol{x}) \mid \boldsymbol{x} \in \mathcal{X}\}. \tag{17}$$

In addition, for some initial state $\rho_0$, we can define an ensemble of the data-embedded quantum states $\mathbb{S}_{\boldsymbol{x}} = \{U(\boldsymbol{x})\rho_0 U^\dagger(\boldsymbol{x})\}$ for all $U(\boldsymbol{x}) \in \mathbb{U}_{\boldsymbol{x}}$. Consequently, performing an average over all the input data is equivalent to an average over the ensemble of data-encoded unitaries $\mathbb{U}_{\boldsymbol{x}}$, or the data encoded states $\mathbb{S}_{\boldsymbol{x}}$.

More concretely, we can measure the expressivity of a given ensemble $\mathbb{U}$ by how close it is from a 2-design (a pseudo-random distribution that agrees with the random distribution up to the second moment). Adopting the definition of expressivity used in refs. 25,66,67, the following superoperator formally quantifies the

distance between $\mathbb{U}$ and an ensemble that forms a 2-design,

$$\mathcal{A}_{\mathbb{U}}(\cdot) := \mathcal{V}_{\text{Haar}}(\cdot) - \int_{\mathbb{U}} dU U^{\otimes 2}(\cdot)(U^\dagger)^{\otimes 2}. \tag{18}$$

Here $\mathcal{V}_{\text{Haar}}(\cdot) = \int_{\mathcal{U}(d)} d\mu(V) V^{\otimes 2}(\cdot)(V^\dagger)^{\otimes 2}$ is an integral over Haar ensemble and the second term is an integral over $\mathbb{U}$. In our case, we have the data-encoded ensemble as our ensemble of interest i.e., $\mathbb{U} = \mathbb{U}_{\boldsymbol{x}}$. Given an input state $\rho_0$, the trace norm

$$\varepsilon_{\mathbb{U}_{\boldsymbol{x}}} := \|\mathcal{A}_{\mathbb{U}_{\boldsymbol{x}}}(\rho_0)\|_1, \tag{19}$$

can be chosen as a data-dependent expressivity measure. The data-dependence of $\varepsilon_{\mathbb{U}_{\boldsymbol{x}}}$ stems from the dependence of $\mathbb{U}_{\boldsymbol{x}}$, Eq. (17), on the input data $\mathcal{X}$. Thus $\varepsilon_{\mathbb{U}_{\boldsymbol{x}}}$ takes into account not just the expressivity of the embedding but also the randomness of the input dataset. The measure equals zero, $\varepsilon_{\mathbb{U}_{\boldsymbol{x}}} = 0$, only when $\mathbb{U}_{\boldsymbol{x}}$ is maximally expressive (i.e., when it agrees with the uniform distribution up to at least the second moment).

To understand why expressivity can be an issue for kernel-based methods, let us consider the fidelity quantum kernel of Eq. (3). This kernel requires computing the inner product between two vectors in an exponentially large Hilbert space. As such, for highly expressive embeddings we are essentially evaluating the inner product between two approximately random (and hence orthogonal) vectors, thus leading to typical kernel values being exponentially small. That is, kernel values tend to concentrate with increased expressivity. The following theorem establishes the formal relationship between the expressivity of the unitary embedding and the concentration of quantum kernels.

**Theorem 1.** (Expressivity-induced concentration) Consider the fidelity quantum kernel as defined in Eq. (3) and the projected quantum kernel as defined in Eq. (4). Assume that input data $\boldsymbol{x}$ and $\boldsymbol{x}'$ are drawn from the same distribution, leading to an ensemble of unitaries $\mathbb{U}_{\boldsymbol{x}}$ as defined in Eq. (17). We have

$$\Pr_{\boldsymbol{x},\boldsymbol{x}'}\left[|\kappa(\boldsymbol{x},\boldsymbol{x}') - \mathbb{E}_{\boldsymbol{x},\boldsymbol{x}'}[\kappa(\boldsymbol{x},\boldsymbol{x}')]| \geq \delta\right] \leq \frac{G_n(\varepsilon_{\mathbb{U}_{\boldsymbol{x}}})}{\delta^2}, \tag{20}$$

where $\varepsilon_{\mathbb{U}_{\boldsymbol{x}}} = \| \mathcal{A}_{\mathbb{U}_{\boldsymbol{x}}}(\rho_0)\|_1$ is the data-dependent expressivity measure over $\mathbb{U}_{\boldsymbol{x}}$ defined in Eq. (19), and $G_n(\varepsilon_{\mathbb{U}_{\boldsymbol{x}}})$ is a function of $\varepsilon_{\mathbb{U}_{\boldsymbol{x}}}$ defined as below.

1. For the fidelity quantum kernel $\kappa(\boldsymbol{x},\boldsymbol{x}') = \kappa^{FQ}(\boldsymbol{x},\boldsymbol{x}')$, we have

$$G_n(\varepsilon_{\mathbb{U}_{\boldsymbol{x}}}) = \beta_{\text{Haar}} + \varepsilon_{\mathbb{U}_{\boldsymbol{x}}}\left(\varepsilon_{\mathbb{U}_{\boldsymbol{x}}} + 2\sqrt{\beta_{\text{Haar}}}\right), \tag{21}$$

where $\beta_{\text{Haar}} = \frac{1}{2^{n-1}(2^n+1)}$

2. For the projected quantum kernel $\kappa(\boldsymbol{x},\boldsymbol{x}') = \kappa^{PQ}(\boldsymbol{x},\boldsymbol{x}')$, we have

$$G_n(\varepsilon_{\mathbb{U}_{\boldsymbol{x}}}) = 4\gamma n\left(\tilde{\beta}_{\text{Haar}} + \varepsilon_{\mathbb{U}_{\boldsymbol{x}}}\right), \tag{22}$$

where $\tilde{\beta}_{\text{Haar}} = \frac{3}{2^{n+1}+2}$.

Theorem 1 establishes that higher embedding expressivity leads to greater quantum kernel concentration. That is, the upper bound on the kernel concentration becomes smaller when $U(\boldsymbol{x})$ is more expressive. In the limit where $\mathbb{U}_{\boldsymbol{x}}$ forms an ensemble that is exponentially close to a 2-design (corresponding to $\varepsilon_{\mathbb{U}_{\boldsymbol{x}}} \in \mathcal{O}(1/b^n)$ for $b > 1$), the kernel exponentially concentrates, and so exponentially many measurement shots are required to evaluate the kernel on a quantum device. Note that the fidelity kernel exponentially concentrates to some exponentially small value i.e., $\mu = \mathbb{E}_{\boldsymbol{x},\boldsymbol{x}' \sim \text{Haar}}[\kappa^{FQ}(\boldsymbol{x},\boldsymbol{x}')] = 1/2^n$.

We stress that the proof of Theorem 1 makes no assumptions on the form of $U(\boldsymbol{x})$. This means the theorem holds for a wide range of embedding architectures, including both problem-agnostic[11,15,29,37,46,68] and problem-inspired embeddings[6–8]. In Supplementary Note IV A, we generalize Theorem 1 by relaxing the assumption that $\boldsymbol{x}$ and $\boldsymbol{x}'$ are drawn from the same distribution. This is relevant, for example, in binary classification tasks where one might want to analyze the behavior of the kernel when a pair of inputs are drawn from different training ensembles. Here, we find a similar conclusion, where higher expressivity leads to more concentrated kernel values.

Although Theorem 1 is stated in terms of the unitary embedding of classical data, the theorem is also applicable to quantum data. That is, it can also be applied when the input data is a collection of pure quantum states generated directly from some quantum process of interest. This follows from the fact that given a set of quantum data states, there is a (potentially unknown) underlying ensemble of unitaries associated with preparing this set of states. Since we can associate each of these unitaries with a classical label, we can associate the quantum data with an encoding ensemble $\mathbb{U}_{\boldsymbol{x}}$ and apply Theorem 1 as before. For example, consider a Hamiltonian $H(\boldsymbol{x})$ where $\boldsymbol{x}$ are parameters describing the Hamiltonian (e.g. perhaps on-site energies or interaction strengths). The quantum data generated by an evolution under $H(\boldsymbol{x})$ for time $T$ can be expressed as $\{U(\boldsymbol{x}_i)|0\rangle = e^{-iH(\boldsymbol{x}_i)T}|0\rangle\}_i$.

We numerically probe the dependence of the concentration of quantum kernel values on the expressivity of the data embedding. To do so, we consider a Hardware Efficient Embedding (HEE)[29], comprised of $L$ layers of data-dependent single-qubit rotations around the $x$-axis followed by entangling gates (Fig. 4). We further consider a data re-uploading strategy where an input data point is repeatedly uploaded into the data embedding[15,29,69,70]. In particular, the $i^{\text{th}}$-component of a data point $\boldsymbol{x}$ is encoded as the rotation angle of qubit $i$ in every HEE layer.

We chose to focus on the binary classification task of distinguishing handwritten '0' and '1' digits from the MNIST dataset[71]. As sketched in Fig. 5a, each individual image (i.e., an input data point) is dimensionally reduced to a real-valued vector of length $n$ using principle component analysis. We refer the reader to Appendix F of ref. 29 for more details.

For a dataset $\mathcal{S} = \{\boldsymbol{x}_i, y_i\}_{i=1}^{N_s}$ of size $N_s$, we evaluate the kernel values over all possible different pairs of inputs in $\mathcal{S}$. Thus we consider the set of values: $\mathcal{K}_\mathcal{S} = \{\kappa(\boldsymbol{x}_1, \boldsymbol{x}_2), \kappa(\boldsymbol{x}_1, \boldsymbol{x}_3), \ldots, \kappa(\boldsymbol{x}_{N_s-1}, \boldsymbol{x}_{N_s})\}$. We note that kernel values for pairs of identical inputs are always 1 and are so excluded from $\mathcal{K}_\mathcal{S}$. To study the degree to which the quantum kernels probabilistically concentrate, we compute the variance $\text{Var}_{\boldsymbol{x},\boldsymbol{x}'}[\kappa(\boldsymbol{x},\boldsymbol{x}')]$ over $\mathcal{K}_\mathcal{S}$.

Figure 6 shows results for the scaling of the kernel variance as a function of the number of qubits $n$ and HEE layers $L$. As $L$ increases, the expressivity of the ansatz increases, and for sufficiently large $L$ we observe exponential concentration of both the fidelity and projected quantum kernels. We note that while the projected quantum kernel

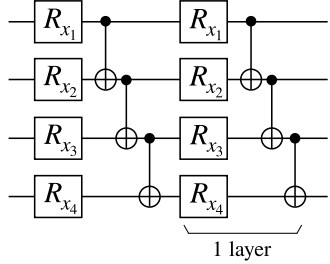

**Fig. 4 | Hardware Efficient Embedding (HEE).** A layer is composed of single qubit x-rotations where the rotation angle on qubit $k$ is given by the $k_{\text{th}}$ component of the input data point $\boldsymbol{x}$. After each layer of rotations, one applies entangling gates acting on adjacent pairs of qubits.

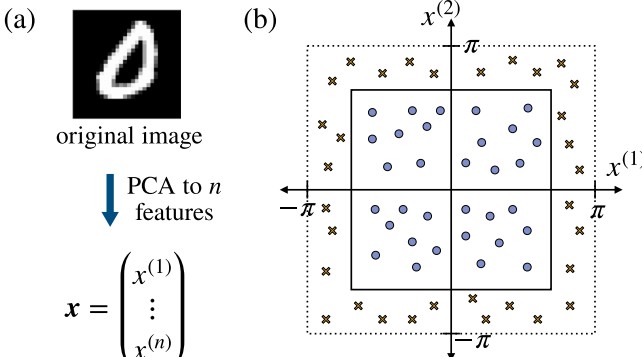

**Fig. 5 | Datasets. a** An input data point $\boldsymbol{x}$ is obtained from dimensionally reducing an original MNIST image to $n$ features using principal component analysis. We assign label $-1$ if the original image is digit '0' and 1 if the original image is digit '1'. **b** A hypercube of width $2\pi/2^{1/n}$ is centered at the origin. An input data point $\boldsymbol{x}$ with each of its component bounded between $-\pi$ and $\pi$ has an associated label $y = 1$ if the point is inside the hypercube (represented by a circle) and $y = -1$, otherwise (represented by a cross).

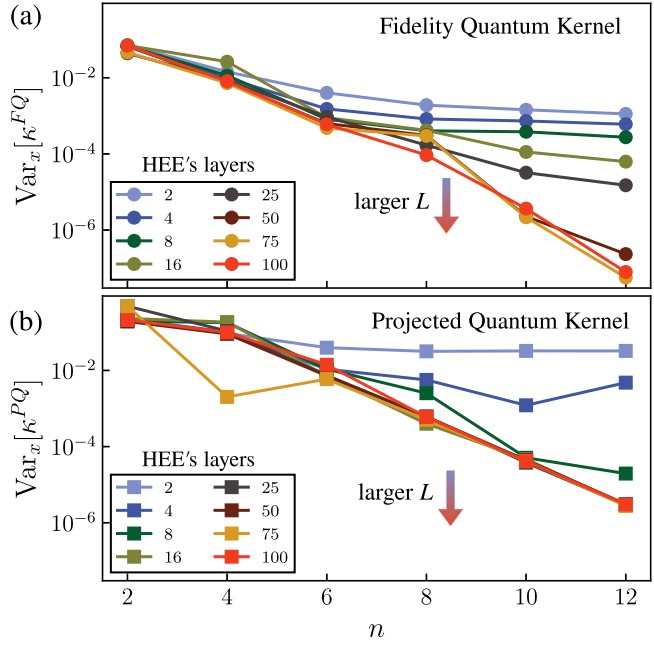

**Fig. 6 | Effect of expressivity on quantum kernels.** We plot variances of the (**a**) fidelity and (**b**) projected quantum kernels, as a function of $n$ and $L$. The classical data from the MNIST dataset ($N_s = 40$) is encoded via an $L$-layer HEE.

reaches the exponential decay regime at shorter depths (i.e., roughly $L \geqslant 16$ for the projected kernel, compared to $L \geqslant 75$ for the fidelity kernel), we generally observe smaller variances (and so stronger concentration) for the fidelity kernel than the projected kernel.

Theorem 1 and the numerics presented in Fig. 6 highlight the importance of the expressivity of quantum kernels. Namely, highly expressive encodings (whether using fidelity or projected kernels) should be avoided. Or, more concretely, unstructured data-embeddings[11,15,46,68] should generally be avoided and the data structure should be taken into account when designing a data-embedding (for instance by constructing geometrically inspired embedding schemes[38–41]).

**2. Entanglement-induced concentration.** In the previous section we saw that high expressivity can be an issue due to the fact that kernels

(such as the fidelity kernel) compare inner products of objects in exponentially large spaces. This issue can be mitigated using projected kernels, which reduce the dimension of the feature space. However, a different issue arises here due to the non-local correlations between the qubits. Namely, the entanglement of the encoded state is another potential source of concentration. Intuitively, this follows from the fact that tracing out qubits in very entangled encoded states, leads to local states that are close to maximally mixed.

**Theorem 2.** (Entanglement-induced concentration) Consider the projected quantum kernel as defined in Eq. (4). For a given pair of data-encoded states associated with $\boldsymbol{x}$ and $\boldsymbol{x}'$, we have

$$|1 - \kappa^{PQ}(\boldsymbol{x},\boldsymbol{x}')| \leqslant (2 \ln 2)\gamma \Gamma_s(\boldsymbol{x},\boldsymbol{x}'), \tag{23}$$

where

$$\Gamma_s(\boldsymbol{x},\boldsymbol{x}') = \sum_{k=1}^{n} \left[ \sqrt{S\left(\rho_k(\boldsymbol{x}) \middle\| \frac{\mathbb{1}_k}{2}\right)} + \sqrt{S\left(\rho_k(\boldsymbol{x}') \middle\| \frac{\mathbb{1}_k}{2}\right)} \right]^2, \tag{24}$$

where we denote $S(\cdot \| \cdot)$ as the quantum relative entropy, $\rho_k$ as a reduced state on qubit $k$, and $\mathbb{1}_k$ as the maximally mixed state on qubit $k$.

Theorem 2 upper bounds the deviation of kernel values from a fixed value of 1 with the relative entropy between the reduced states of the encoded data and a maximally mixed state of a single qubit. In addition, unlike the results in the previous sections, the exponential concentration bounds here are deterministic. In the case where the entanglement of the encoded states obeys a volume law, that is $S(\rho_k(\boldsymbol{x}) \| \frac{\mathbb{1}_k}{2}), S(\rho_k(\boldsymbol{x}') \| \frac{\mathbb{1}_k}{2}) \in \mathcal{O}(1/2^{n-1})$ for all subsystems, the kernel values deterministically exponentially concentrate to 1. For encoded states that obey an area-law scaling, i.e. $S(\rho_k(\boldsymbol{x}) \| \frac{\mathbb{1}_k}{2}), S(\rho_k(\boldsymbol{x}') \| \frac{\mathbb{1}_k}{2}) \in \mathcal{O}(1)$ for all subsystems, the story is more complex. Theorem 2, as an upper bound, allows (but does not guarantee) that such data states do not concentrate exponentially.

It is worth highlighting that the entanglement-induced bound in Theorem 2 is stated for a given pair of data-encoded states, and not as an average over all possible data pairs. Hence, it is thus natural to determine classes of data and embeddings where concentration will arise with high probability, e.g., cases when the encoded states obey a volume law of entanglement. First, we note that if the ensemble of encoded data states forms at least a 4-design, then most of the encoded states to obey a volume-law scaling[72,73]. However, in this case, our bound on expressivity already implies that the kernel's exponentially concentrate so the entanglement-induced result is redundant.

Entanglement-induced concentration can also occur in cases where the embedding is not highly expressive but still leads to states satisfying a volume-law. Here, Theorem 2 implies that the kernel values of the projected quantum kernels will exponentially concentrate. In this case performing any supervised learning task with the projected quantum kernels will fail with a polynomial number of measurement shots. As an example, consider binary classification and assume that one manages to construct a $U(\boldsymbol{x})$ that maps the input data into one of the two orthogonal sets of volume-law entangled states depending on the true label of the input. In this setting, the trained model with the fidelity kernel should not face issues associated with exponential concentration. However, if we use the projected quantum kernel, we cannot perform the task better than random guessing without spending an exponential number of shots. This statement is formalized in the following corollary.

**Corollary 2.** Consider the projected quantum kernel as defined in Eq. (4). If all the states in the ensemble $\mathbb{S}_{\text{train}}$ generated from the training

dataset obey volume law scaling, we have

$$|1 - \kappa^{PQ}(\boldsymbol{x},\boldsymbol{x}')| \in \mathcal{O}(n2^{-n}), \tag{25}$$

for all $\boldsymbol{x}$ and $\boldsymbol{x}'$ in the training data.

Thus, when using projected kernels, highly entangling encodings should be avoided to ensure predictability on unseen data. We note that fidelity kernels (with pure input states) are not affected by entanglement in this manner as they do not require tracing qubit out. Lastly, we stress that Theorem 2 and Corollary 2 are readily applied to quantum data. Indeed, entanglement-induced concentration may well be more problematic in this case since if the quantum data is already highly entangled then there is little that one can do. (In contrast, for classical data one may simply avoid highly entangled embeddings). As an example, consider a quantum dataset generated by evolving different initial states with either $U_1$ or $U_2$ where $U_1$ and $U_2$ are unitaries drawn from the Haar measure over the unitary group. Since Haar random evolution leads to a volume-law scaling, classifying whether a given state is evolved by $U_1$ or $U_2$ cannot be done efficiently using projected quantum kernels.

**3. Global-measurement-induced concentration.** Global measurements can be another source of exponential concentration. A global measurement is a measurement that acts non-trivially on all $n$ qubits. Such global measurements are required by design to compute fidelity kernels but not projected kernels. In broad terms global measurements can lead to concentration because we are attempting to extract global information about a state that lives in an exponentially large Hilbert space. While projected quantum kernels do not face these difficulties due to their local construction, we argue that global measurements can lead to problems for the fidelity kernel.

To illustrate this problem, we provide an example where the data embedding has low expressivity and contains no entanglement and yet it is still possible to have exponential concentration. Consider the tensor product unitary data embedding $U(\boldsymbol{x}) = \bigotimes_{k=1}^{n} U_k(x_k)$ with $x_k$ being a $k$-th component of $\boldsymbol{x}$, and $U_k$ being a single-qubit rotation about the $y$-axis on the $k$-th qubit. The following proposition holds.

**Proposition 3.** (Global-measurement-induced concentration) Consider the fidelity quantum kernel as defined in Eq. (3) where the data embedding is of the form $U(\boldsymbol{x}) = \bigotimes_{k=1}^{n} U_k(x_k)$ with $x_k$ being an input component encoded in the qubit $k$, and $U_k$ being a single-qubit rotation about the $y$-axis on the $k$-th qubit. For an input data point $\boldsymbol{x}$, assume that all components of $\boldsymbol{x}$ are independent and uniformly sampled in $[-\pi, \pi]$. Given a product initial state $\rho_0 = \bigotimes_{k=1}^{n}|0\rangle\langle0|$, we have,

$$\Pr_{\boldsymbol{x},\boldsymbol{x}'}\left[|\kappa^{FQ}(\boldsymbol{x},\boldsymbol{x}') - 1/2^n| \geq \delta\right] \leqslant \left(\frac{3}{8}\right)^n \cdot \frac{1}{\delta^2}. \tag{26}$$

Intuitively, the result in Proposition 3 can be understood as following from the fact that the fidelity between two product states is usually exponentially small. In Supplementary Note VI A, we further generalize this proposition to the case when $U_k$ is a general unitary, which also leads to a concentration result.

We remark that the assumptions underlying Proposition 3 can be relevant in practice. For example, consider classifying whether or not a given point $\boldsymbol{x}$ in $n-$dimensional space (with each component bounded between $[-\pi, \pi]$) stays inside a hypercube centered at the origin with the width of $\frac{2\pi}{2^{1/n}}$ (see Fig. 5(b)). Note that the width of the hypercube is chosen so there is a 0.5 probability of a randomly chosen point being in or out of the hypercube. For this task, an individual data point in the training dataset is generated by uniformly drawing each vector component from the range $[-\pi, \pi]$. Since here data points are obtained via

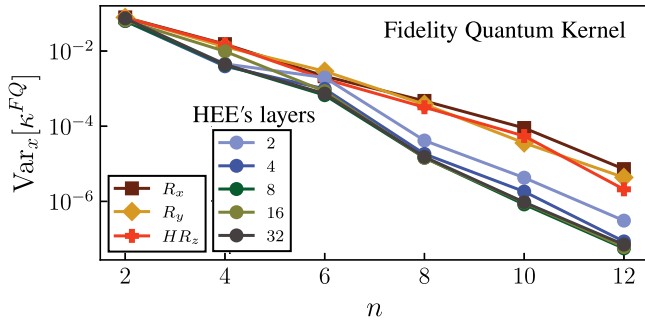

**Fig. 7 | Global-measurement concentration of quantum kernels.** We plot the variance of the fidelity kernel as a function of $n$ using different data-embeddings, namely a single layer of one qubit rotations ($R_x$, $R_y$, Hadamard followed by $R_z$) and HEE with $L$ layers. The components of $N_s = 40$ input data points are independent and uniformly drawn from $[-\pi, \pi]$.

uniformly sampling each component independently, the above assumptions are satisfied.

We numerically study the concentration of the kernels for this classification task in Fig. 7. To reduce the effects of expressivity and entanglement, we first select the data embedding to be a single layer of one qubit rotations ($R_x$, $R_y$, Hadamard followed by $R_z$). Similar to the HEE, each component of an individual data point is embedded as a rotation angle. We observe an exponential decay in the variance of the fidelity kernel in a good agreement with our theoretical predictions.

While Proposition 3 is derived with a tensor product embedding, similar results are expected when dealing with more general unstructured embeddings such as hardware efficient embeddings. This is because the additional complexity from using an unstructured embedding can only increase the kernel concentration (due to increased expressivity and entanglement). This is highlighted in Fig. 7 where we additionally consider an $L$-layered HEE and see that increasing expressivity can accelerate the exponential decay.

Nonetheless, it is important to stress that global measurements do not always lead to exponential concentration. For example, if the encoded quantum states are not too far away in Hilbert space, such that the fidelity kernel values concentrate no worse than polynomially in $n$, their overlap can be efficiently resolved. For example, the MNIST classification task does not satisfy the assumptions of Proposition 3. As a result, as shown in Fig. 6, global measurements do not lead to the exponential concentration of the fidelity kernel for low depth ansatze. This demonstrates that the structure of the training data matters and global measurements do not always lead to exponential concentration.

Thus, the key message here is that when using global measurements to evaluate the kernel, the embedding must be chosen particularly carefully such that the fidelity between any pair of encoded quantum states is at least in $\Omega(1/\text{poly}(n))$. To achieve this, one can either take the problem's structure into consideration when building the embedding[8,38,39,42] or further reduce the expressivity of problem-agnostic embeddings[43].

**4. Noise-induced concentration.** Hardware noise may disrupt and destroy information in the encoded quantum states, providing another source of concentration. To analyze the effect of noise, we here further suppose the data-embedding can be decomposed into $L$ layers of data-encoding unitaries

$$U(\boldsymbol{x}) = \prod_{l=1}^{L} U_l(\boldsymbol{x}_l) \qquad (27)$$

where $\boldsymbol{x}_l$ is an input associated with $\boldsymbol{x}$ that is encoded in the layer $l$. We remark that from our construction, $\boldsymbol{x}_l$ can be either the $l$ th component of the input data $\boldsymbol{x}$ or a fixed vector $\boldsymbol{x}$ that is encoded

repeatedly. Although the form of the data embedding is slightly less general than the one described in the noiseless sections, it still covers a large class of data embedding ansatze including the Hardware Efficient Embedding (HEE)[11,15,29,37,46,68], the Quantum Alternative Operator Ansatz (QAOA)[74], the Hamiltonian Variational Embedding (HVE)[6,43] and Instantaneous Quantum Polynomial (IQP) embedding[29,37,43].

We model the hardware noise as a Pauli noise channel applied before and after every layer of the embedding, similar to the model considered in ref. 28. The output state of the noisy embedding circuit is given by

$$\tilde{\rho}(\boldsymbol{x}) = \mathcal{N} \circ \mathcal{U}_L(\boldsymbol{x}_L) \circ \mathcal{N} \circ \ldots \circ \mathcal{N} \circ \mathcal{U}_1(\boldsymbol{x}_1) \circ \mathcal{N}(\rho_0) \qquad (28)$$

where $\mathcal{U}_l(\boldsymbol{x}_l)$ is the channel corresponding to the unitary $U_l(\boldsymbol{x}_l)$ and $\mathcal{N} = \mathcal{N}_1 \otimes \ldots \otimes \mathcal{N}_n$ is a local Pauli noise channel. Specifically, in this work we consider unital channels such that the effect of $\mathcal{N}_j$ on each local Pauli operator $\sigma \in \{X, Y, Z\}$ is given by

$$\mathcal{N}_j(\sigma) = q_\sigma \sigma, \qquad (29)$$

where $-1 < q_\sigma < 1$. We remark that the noiseless regime corresponds to $q_\sigma = 1$ for all qubits. The strength of the noise can be quantified by a characteristic noise parameter which is defined as

$$q = \max\{|q_X|, |q_Y|, |q_Z|\}. \qquad (30)$$

The following theorem summarizes the impact of noise on quantum kernels.

**Theorem 3.** (Noise-induced concentration) Consider the $L$-layered data embedding circuit defined in Eq. (27) with input state $\rho_0$ and the layerwise Pauli noise model defined in Eq. (28) with characteristic noise parameter $q < 1$. The concentration of quantum kernel values may be bounded as follows

$$|\tilde{\kappa}(\boldsymbol{x}, \boldsymbol{x}') - \mu| \leqslant F(q, L). \qquad (31)$$

1. For the fidelity quantum kernel $\tilde{\kappa}(\boldsymbol{x}, \boldsymbol{x}') = \tilde{\kappa}^{FQ}(\boldsymbol{x}, \boldsymbol{x}')$, we have $\mu = 1/2^n$, and

$$F(q, L) = q^{2L+1} \left\| \rho_0 - \frac{\mathbb{1}}{2^n} \right\|_2. \qquad (32)$$

2. For the projected quantum kernel $\tilde{\kappa}(\boldsymbol{x}, \boldsymbol{x}') = \tilde{\kappa}^{PQ}(\boldsymbol{x}, \boldsymbol{x}')$, we have $\mu = 1$, and

$$F(q, L) = (8 \ln 2)\gamma n q^{b(L+1)} S_2 \left( \rho_0 \left\| \frac{\mathbb{1}}{2^n} \right. \right), \qquad (33)$$

where $S_2(\cdot \| \cdot)$ denotes the sandwiched 2-Rényi relative entropy and $b = 1/(2 \ln(2)) \approx 0.72$.

Additionally, the noisy data-encoded quantum state $\tilde{\rho}(\boldsymbol{x})$ concentrates towards the maximally mixed state as

$$\left\| \tilde{\rho}(\boldsymbol{x}) - \frac{\mathbb{1}}{2^n} \right\|_2 \leqslant q^{L+1} \left\| \rho_0 - \frac{\mathbb{1}}{2^n} \right\|_2. \qquad (34)$$

Theorem 3 shows that the concentration of quantum kernels due to noise is exponential in the number of layers $L$ for both the fidelity and projected quantum kernels. This is a consequence of the encoded state concentrating towards the maximally mixed state, as captured in Eq. (34). In addition, we note that the noise-induced concentration

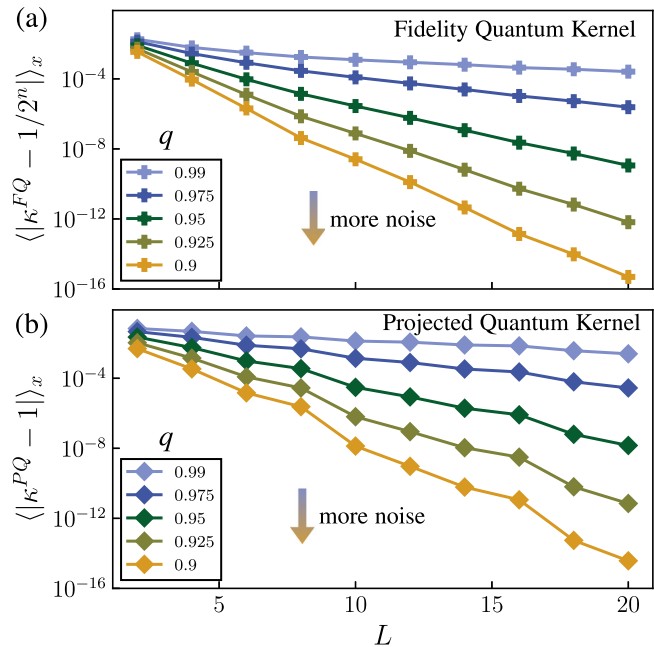

**Fig. 8 | Effect of noise.** We plot the average of the difference between the quantum kernels and their respective fixed point $\mu$ over different input data points and for different number of layers $L$ and noise parameter $q$. We consider the fidelity quantum kernel in **a** with $\mu = 1/2^n$ and the projected quantum kernel in **b** with $\mu = 1$. We use the MNIST dataset with $N_s = 40$ and $n = 8$.

bounds here are deterministic due to the noise acting independently of the input data.

If quantum kernel-based methods are to provide any quantum advantage, the data embedding part must be hard to classically simulate. For example, when using embeddings with local connectivity, we are largely interested in the regime of moderately deep circuits where $L$ scales at least linearly in $n$[37]. However, it is precisely this regime in which our bounds suggest kernels will exponentially concentrate due to an effect of noise. In particular, when the number of layers $L$ scales polynomially with the number of qubits $n$, $F(q, L)$ decays exponentially in the number of qubits. We stress that the exponential decay nature of the concentration bounds persists for all $q < 1$ and different values of noise characteristics only lead to different exponential decay rates.

The impact of noise on the concentration of kernels is studied in Fig. 8 where we plot the average of $|\kappa^{FQ}(x, x') - 1/2^n|$ and $|\kappa^{PQ}(x, x') - 1|$ for the MNIST dataset as a function of the depth $L$ of the HEE embedding and the noise characteristic $q$. We observe exponential concentration with $L$, with the concentration stronger for higher noise levels $q$, in agreement with Theorem 3. We note that in our numerical simulations, noise only acts before and after single-qubit gates and we assume noiseless implementations of entangling gates. Therefore, in real experiments, where gate fidelity of entangling gates is generally worse than single-qubit gates, we expect the noise to dominate at a faster pace.

In Supplementary Note VIII, we argue that, similar to noise-induced BPs in VQAs[32,75,76], the exponential concentration cannot be resolved with current common error mitigation techniques including Zero-Noise Extrapolation[77–80], Clifford Data Regression[81], Virtual Distillation[82,83] and Probabilistic Error Cancellation[78,79]. Hence, noise-induced concentration results poses a significant barrier to the successful implementation of quantum kernel methods on near term hardware.

### Training parameterized quantum kernels
Given the problems associated with expressivity-induced concentration, it is generally advisable to avoid problem-agnostic embeddings

and instead try and take advantage of the data structure of the problem. However, in many cases, constructing such problem-inspired embeddings is highly non-trivial. An alternative is to allow the data embedding itself to be parametrized and then train the embedding. Such strategies have been shown to improve generalization of the kernel-based quantum model[68,74]. We note that this is an additional process to train and select an appropriate embedding before implementing the standard quantum kernel algorithm (with this selected embedding).

Here we consider a parametrized data embedding $U(x, \theta)$, where $\theta$ is a vector of trainable parameters (typically corresponding to single qubit rotation angles). For a given input data vector $x$, an ensemble of data embedding unitaries can be generated by varying the parameters $\theta$. This in turn generates a family of parametrized quantum kernels $\kappa_\theta(x, x')$. Let $\theta = \theta^*$ be the optimal parameters found by training the embedding. The optimally embedded kernel now corresponds to $\kappa(x, x') = \kappa_{\theta^*}(x, x')$ and the remaining process to obtain the optimal model is the same as that described in Section IA.

The standard approach to obtain the optimal kernel is to train the parameters $\theta^*$ via standard optimization techniques[18,68,74], which in turn requires defining a loss function one needs to minimize. For instance, in a binary classification task where the true labels are either +1 or −1, the ideal kernel is +1 if the input data are in the same class and is −1 otherwise. In practice, however, one can only approximate the ideal kernel as

$$\kappa_{\text{ideal}}(x_i, x_j) = y_i y_j, \qquad (35)$$

using the given training data $\mathcal{S}$. The kernel target alignment measures the similarity between the parameterized kernel and the approximated ideal kernel[68,84]

$$TA(\theta) = \frac{\sum_{i,j} y_i y_j \kappa_\theta(x_i, x_j)}{\sqrt{\left(\sum_{i,j} (\kappa_\theta(x_i, x_j))^2\right)\left(\sum_{i,j} (y_i y_j)^2\right)}}. \qquad (36)$$

As minimizing the target alignment corresponds to aligning the parametrized kernel to the ideal kernel, we can use $TA(\theta)$ as a loss function. Crucially, unlike the training of the model itself, the associated loss function for training the embedding is generally non-convex.

Training the parameterized data-embedding $U(x, \theta)$ has been recently proposed as an approach to improve generalization quantum kernel-based methods[68,74]. In particular, ref. 68 showed that optimizing the kernel target alignment $TA(\theta)$ of Eq. (36) leads to data-embedding schemes with better performance than unstructured embeddings for various MNIST-based binary classification tasks. However, this assumes that one can successfully train the target alignment.

Here we study the trainability of $TA(\theta)$. Namely, we discuss what features of the parameterized embedding $U(x, \theta)$ can lead to exponential concentration and therefore to exponentially flat parameter landscapes (i.e., a BP). First, we show that the variance of $TA(\theta)$ with respect to the variational parameters $\theta$ is upper bounded by the variances of the parameterized quantum kernels $\kappa_\theta(x, x')$

**Proposition 4.** (Concentration of kernel target alignment) Consider an arbitrary parameterized kernel $\kappa_\theta(x, x')$ and a training dataset $\{x_i, y_i\}_{i=1}^{N_s}$ for binary classification with $y_i = \pm 1$. The probability that the kernel target alignment $TA(\theta)$ (defined in Eq. (36)) deviates from its mean value is approximately bounded as

$$\Pr_\theta\left[|TA(\theta) - \mathbb{E}_\theta[TA(\theta)]| \geq \delta\right] \lesssim \frac{M \sum_{i,j} \text{Var}_\theta[\kappa_\theta(x_i, x_j)]}{\delta^2}, \qquad (37)$$

with $M = \frac{8 + N_s^3(9(N_s-1)^2 + 16)}{4N_s}$.

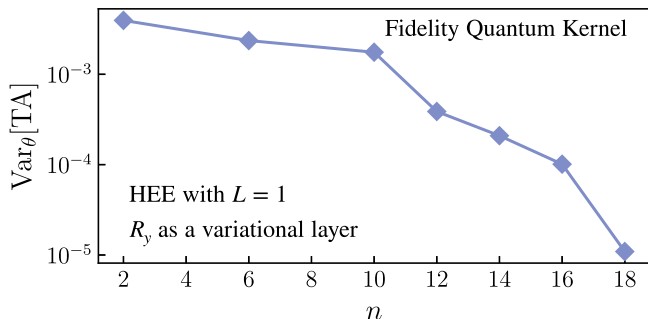

**Fig. 9 | Kernel target alignment.** The variance of TA($\theta$) with respect to variational parameters is plotted as a function of $n$. Here we use the hypercube dataset with $N_s = 10$.

If $\text{Var}_\theta[\kappa_\theta(x_i, x_j)]$ vanishes exponentially in the number of qubits for all pairs in the training data, the probability that TA($\theta$) deviates by an amount $\delta$ from its mean vanishes exponentially with the size of the problem. In this case, the parameter landscape of TA($\theta$) becomes exponentially flat and hence TA($\theta$) is untrainable with a polynomial number of measurement shots.

In Supplementary Note X, we analyze features leading to exponentially vanishing variances $\text{Var}_\theta[\kappa_\theta(x_i, x_j)]$ and find that the same ones that lead to BPs for QNNs lead to BPs here. Namely, features that are deemed detrimental for trainability in QNNs such as deep unstructured circuits[20,25] and global measurements[21] also lead to BPs here. Thus these features should be avoided when designing parameterized data embeddings for quantum kernels.

We numerically demonstrate the effect of global measurements on training an embedding in Fig. 9. The data embedding consists of a single layer of parameterized single-qubit rotations around $y$-axis followed by a single layer of HEE. We study the variance of the kernel target alignment TA($\theta$) (which determines the flatness of the training landscape[20,21]) for 500 random initialization of the parameters $\theta$. As expected, since the parametrized block acts globally on all qubits, TA($\theta$) exponentially concentrates when one averages over the trainable parameters $\theta$.

## Discussion

Quantum kernels stand out as a promising candidate for achieving a practical quantum advantage in data analysis. This is in part due to the common belief that the optimal quantum kernel-based model can always be obtained[14–17] due to the convexity of the problem. Although this is true, provided that the kernel values can be efficiently obtained to a sufficiently high precision, here we show that there exist scenarios where quantum kernels are exponentially concentrated towards some fixed value and so exponential resources are required to accurately estimate the kernel values. With only a polynomial number of shots, the predictions of the trained model become insensitive to input data and the model performs trivially on unseen data, that is, generalizes poorly. Crucially, in this context generalization cannot be improved by training on more input data points but rather by increasing the number of measurement shots (or using a more appropriate embedding). It is worth stressing that as we assume very little on the form of the data embedding $U(x)$, our analytical bounds hold for a wide range of embedding architectures and schemes, including both problem-agnostic and problem-inspired embeddings.

Our results highlight four aspects to carefully consider when choosing a data embedding for quantum kernels. While much of the literature currently focuses on using problem-agnostic quantum embeddings for quantum kernels[11,15,46,68]; these are typically highly expressive and as such should generally be avoided. Entanglement can also be detrimental when combined with local quantum kernels such as the projected quantum kernels, and suggests that one should be

mindful about using embeddings leading to states satisfying volume-laws of entanglement. Our results on global measurements demonstrate that the fidelity kernel can exponentially concentrate even with a simple embedding that has low expressivity and no entanglement. Consequently, the fidelity kernel should only be used for datasets where the data-embedded states are 'not too distant' in the Hilbert space. Finally, our study of noise suggests that polynomial-depth data embeddings in noisy hardware suffer from exponential concentration, thus presenting a serious barrier to achieve a meaningful quantum advantage in the near term.

In addition, we show that training parametrized quantum kernels using kernel target alignment suffers from an exponentially flat training landscape under similar conditions to those leading to barren plateaus in QNNs. That is, when constructing the parametrized part of the data embeddings, one should avoid features that induce BPs as QNNs such as global measurements and deep unstructured circuits.

Our work provides a systematic study of the barriers to the successful scaling up of quantum kernel methods posed by exponential concentration. Prior work on BPs motivated the community to search for ways to avoid or mitigate BPs such as employing correlated parameters[85] using tools from quantum optimal control[22,86], or developing the field of geometrical quantum machine learning[38–41]. In a similar manner, we stress our results should not be understood as condemning quantum kernel methods, but rather a prompt to develop exponential-concentration-free embeddings for quantum kernels. Crucially, incorporating quantum aspects to machine learning does not always lead to better performance. Indeed, often it will only worsen the performance of the learning models. In particular, if one remains restricted to mimicking the classical techniques without carefully taking into account quantum phenomena, it is unlikely that one will achieve a quantum advantage. Hence distinctly quantum approaches, using specialized quantum structures/symmetries, may prove to be the way forward[37,42].

## Data availability

Data generated and analyzed during the current study are available at the following GitHub repository: https://github.com/Supanut-Thanasilp/Exponential-concentration-in-quantum-kernel-methods.

## Code availability

Code used for the current study are available at the following GitHub repository: https://github.com/Supanut-Thanasilp/Exponential-conce ntration-in-quantum-kernel-methods.

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

## Acknowledgements

We thank Jonas M. Kübler for his comments on Appendix A. ST is supported by the National Research Foundation, Prime Minister's Office, Singapore and the Ministry of Education, Singapore under the Research Centres of Excellence programme and subsequent support from the Sandoz Family Foundation-Monique de Meuron program for Academic Promotion. SW is supported by the Samsung GRP grant. M.C. was initially supported by ASC Beyond Moore's Law project at Los Alamos National Laboratory (LANL). This work was also supported by the Quantum Science Center (QSC), a National Quantum Information Science Research Center of the U.S. Department of Energy (DOE). ZH acknowledges initial support from the LANL Mark Kac Fellowship and subsequent support from the Sandoz Family Foundation-Monique de Meuron program for Academic Promotion.

## Author contributions

The project was conceived by S.T., M.C., and Z.H. Theoretical results were proved by S.T., S.W., M.C., and Z.H. Numerical implementations were performed by S.T. The manuscript was written by S.T., S.W., M.C., and Z.H.

## Competing interests

The authors declare no competing interests.
