## [Peer Review File · Nature Communications]

Exponential concentration in quantum kernel methodsREVIEWER COMMENTS

Reviewer #1 (Remarks to the Author):

This paper addresses the problem of assessing the effectiveness of kernel methods in quantum machine learning under the restrictions of the current quantum hardware.

Quantum kernels share with classical kernels some mathematical properties that allow for a theoretical analysis exploiting the optimality guarantees provided by the convexity of the methods. However, as already pointed out in Ref.7 (see also references therein), simply embedding classical data into highly expressive RKHS does not lead to any quantum advantage without the use of some inductive bias which meaningfully exploits knowledge about the data that cannot be expressed classically. Moreover, in the same reference it is established that, similarly to the barren plateau problem for QNN, kernel values tend to exponentially vanish in the number of qubits, implying that exponentially many measurements are necessary to obtain accuracy.

These results are re-established and extended in the manuscript by Thanasilp et al. by introducing the concept of 'exponential concentration' and analysing the characteristics of the embeddings leading to this problem. In all generality, the exponential concentration problem consists in the impossibility of distinguishing a specific result of a quantum observable from its expectation value without an exponential number of measurements. When the quantum observable is a quantum kernel, this situation obviously hinders the trainability of the quantum kernel for supervised learning, as this is based on the relative differences between training vectors (i.e. kernel values).

Thus the authors' study of conditions and criteria for defining 'good' embeddings by avoiding the concentration problem is very appropriate although the results are not so surprising. They formally show some bounds on the expressivity of the embedding (for fidelity and projected kernels), and the degree of entanglement (for projected kernels) in the encoded states. They also show that global measuring is a source of exponential concentration for fidelity kernels, as well as the noise in layered embedding circuits (for both fidelity and projected quantum kernels).

Comments:

- The paper is well-written and the results are sound as far as I could check. (There are typos, e.g.

on pag 2, second column, line 7 from bottom, there is a `of` missing between `study` and `the number`).

- The claim that the issue of the generalisation property of quantum kernels has a wider treatment in the literature than trainability of quantum kernels is not completely true. The references mentioned in the conclusion should include a longer list than just [8,14,42] with a bigger intersection with the references mentioned for the generalisation (e.g. 7).

- The use of `expressibility` is wrong in the context of this paper and, in general, for referring to the ability of expressing or representing something. For this `expressivity` should be used instead. So the power of an ensemble of unitaries to express/represent the unitary group corresponds to its `expressivity`, while its expressibility would refer to the property of being describable/representable/enunciabile /...

In general, the authors make a good job in giving a systematic treatment of the problems in the use of quantum kernels for supervised learning. They should, however, try and reformulate overstatements like `the trainability of quantum kernels has in the large part been overlooked` or `Similar to when BP were first discovered in QNN...`. A more careful reading of the literature show that the results the authors nicely and rigorously show in their paper are not completely new after all.

A consideration comes natural from these results, which holds both for QNN and QK: the use of computational quantum features, especially if limited by hw constraints, is probably not going to give any advantage if restricted to mimicking the (highly specialised and efficient) classical techniques. Maybe the authors should elaborate on this in their conclusion.

Reviewer #2 (Remarks to the Author):

The paper considers the trainability of quantum kernel methods. When given a computable kernel function, kernel methods do not suffer from trainability issues (because the optimal

model can always be found). However, quantum kernels are generally not exactly computable (only approximately computable). The paper shows that the four main obstacles (high expressibility, global measurements, entanglement, and noise) plaguing the trainability of quantum neural networks manifest themselves in quantum kernel methods through the exponential concentration of kernel function values. However, I don't think this implies that the quantum kernel methods cannot learn a model close to optimal is correct (this claim is made in the paper but not proven).

> Strength:

The paper proves that under four circumstances (high expressibility, global measurements, large entanglement, and sufficient noise), the quantum kernel function exponentially concentrates on a useless kernel function (1 on the diagonal and a fixed value μ on the off-diagonal).

This exponential concentration for quantum kernel methods under certain regimes (as identified by the authors) is problematic. The exponential concentration means that the ML model is independent of the data! An ML model that does not depend on data is useless as it would produce the same prediction for essentially all inputs.

The study on when would the exponential concentration arises is very interesting. A circumstance (global measurements) has been identified in a prior work published in Nature Comm [1]. This work identifies three more circumstances that this problem could arise.

> Weakness:

When exponential concentration happens (with off-diagonal values μ not equal to 1), the quantum kernel method could still find an approximately optimal model. When we compare the approximately computed kernel function and the exactly computed kernel function, the corresponding alpha vector will not differ much because the exact kernel matrix is positive definite (with the smallest eigenvalue equal to $1 - \mu$). The positive definiteness ensures that the optimal model does not fluctuate too much with respect to noise in the kernel

entries. Hence the statement that the quantum kernel methods cannot learn a model close to optimal is incorrect.

The statement that it is hard to distinguish between K and K_0 is correct, but that does not imply the trainability issue stated in the work. Because of the above reasoning regarding positive definiteness, I don't think quantum kernel methods have trainability issues (the inability to find a close-to-optimal model) in general (the concentrated value μ is typically close to zero). Exponential concentration is problematic (as stated by the authors), but it does not result in untrainability.

> Overall evaluation:

Because of the incorrect claim, I would not suggest acceptance of the work in its current form. However, the exponential concentration in quantum kernel methods is a serious problem worthy of further emphasis and study. This work provides a solid analysis of this phenomenon. I would consider acceptance if the authors could properly address the incorrect statements in the work.

[1] Huang, Hsin-Yuan, et al. "Power of data in quantum machine learning." *Nature communications* 12.1 (2021): 1-9.

Response to Referee Report of “Exponential concentration and untrainability in quantum kernel methods”

Supanut Thanasilp,^{1,2} Samson Wang,³ M. Cerezo,^{4,5} and Zoë Holmes^{4,2}

¹Centre for Quantum Technologies, National University of Singapore, 3 Science Drive 2 117543, Singapore.

²Institute of Physics, Ecole Polytechnique Fédérale de Lausanne (EPFL), CH-1015 Lausanne, Switzerland

³Imperial College London, London, UK.

⁴Information Sciences, Los Alamos National Laboratory, Los Alamos, NM, USA.

⁵Quantum Science Center, Oak Ridge, TN 37931, USA

I. RESPONSE TO REFEREE 1

This paper addresses the problem of assessing the effectiveness of kernel methods in quantum machine learning under the restrictions of the current quantum hardware. Quantum kernels share with classical kernels some mathematical properties that allow for a theoretical analysis exploiting the optimality guarantees provided by the convexity of the methods. However, as already pointed out in Ref.7 (see also references therein), simply embedding classical data into highly expressive RKHS does not lead to any quantum advantage without the use of some inductive bias which meaningfully exploits knowledge about the data that cannot be expressed classically. Moreover, in the same reference it is established that, similarly to the barren plateau problem for QNN, kernel values tend to exponentially vanish in the number of qubits, implying that exponentially many measurements are necessary to obtain accuracy.

These results are re-established and extended in the manuscript by Thanasilp et al. by introducing the concept of ‘exponential concentration’ and analysing the characteristics of the embeddings leading to this problem. In all generality, the exponential concentration problem consists in the impossibility of distinguishing a specific result of a quantum observable from its expectation value without an exponential number of measurements. When the quantum observable is a quantum kernel, this situation obviously hinders the trainability of the quantum kernel for supervised learning, as this is based on the relative differences between training vectors (i.e. kernel values). Thus the authors’ study of conditions and criteria for defining ‘good’ embeddings by avoiding the concentration problem is very appropriate although the results are not so surprising. They formally show some bounds on the expressivity of the embedding (for fidelity and projected kernels), and the degree of entanglement (for projected kernels) in the encoded states. They also show that global measuring is a source of exponential concentration for fidelity kernels, as well as the noise in layered embedding circuits (for both fidelity and projected quantum kernels).

The paper is well-written and the results are sound as far as I could check. (There are typos, e.g. on pag 2, second column, line 7 from bottom, there is a ‘of’ missing between ‘study’ and ‘the number’).

Response: We thanks the referee for carefully going through our work and are glad that our effort put into preparing the manuscript is appreciated. The summary of our work provided here is accurate.

We agree with the referee that Ref.7 (as well as Ref.6) were the first two works to study exponential concentration in quantum kernels. However, as highlighted by the referee we substantially expand on these works and provide a

systematic analysis of: (i) the sources of kernel concentration, and (ii) the consequences of exponential concentration (see new Appendix C).

To more clearly acknowledge the contribution of prior work and clarify the contribution of our work, we have added Appendix A where we place our results in a broader context. We have got in touch with Jonas M. Kübler who is the first author of Ref.7 to ensure that our understanding of their work is accurate. We also now further stress the role of key prior works in the introduction of the main text.

Changes: The following changes have been made in the new version of the manuscript.

1. We have added a new appendix (i.e., Appendix A.) which discusses the related works and a summary of prior works to the introduction.
2. We have largely modified Section II to properly describe why exponential concentration is problematic with more details discussed in new appendices (i.e., Appendix B. and C).
3. We have corrected the typos highlighted by the referee and proof read the manuscript to fix additional typos.

The use of ‘expressibility’ is wrong in the context of this paper and, in general, for referring to the ability of expressing or representing something. For this ‘expressivity’ should be used instead. So the power of an ensemble of unitaries to express/represent the unitary group corresponds to its ‘expressivity’, while its expressibility would refer to the property of being describable/representable/enunciable /...

Changes: We thank the referee for this useful comment. The use of the word “expressibility” in our work follows the usage of the literature. However, we do agree that expressiveness is more accurate. Hence, we have made these changes throughout the manuscript (from ‘expressibility’ to ‘expressivity’ as well as from ‘expressible’ to ‘expressive’).

The claim that the issue of the generalisation property of quantum kernels has a wider treatment in the literature than trainability of quantum kernels is not completely true. The references mentioned in the conclusion should include a longer list than just [8,14,42] with a bigger intersection with the references mentioned for the generalisation (e.g. 7).

In general, the authors make a good job in giving a systematic treatment of the problems in the use of quantum kernels for supervised learning. They should, however, try and reformulate overstatements like ‘the trainability of quantum kernels has in the large part been overlooked’ or ‘Similar to when BP were first discovered in QNN...’. A more careful reading of the literature show that the results the authors nicely and rigorously show in their paper are not completely new after all.

Response: We thank the reviewer for bringing this to our attention. We have toned down our language through out the manuscript as suggested and, again, to clarify the contribution of our work/more thoroughly acknowledge prior work, we have added a new Appendix A.

Changes:

1. We have softened the language for any potential overstatements.
2. We have added Appendix A and a summary of prior work to the introduction.

A consideration comes natural from these results, which holds both for QNN and QK: the use of computational quantum features, especially if limited by hw constraints, is probably not going to give any advantage if restricted to mimicking the (highly specialised and efficient) classical techniques. Maybe the authors should elaborate on this in their conclusion.

Response/Changes: We thank the reviewer for this helpful suggestion and have now added a comment on this to the discussion section.

II. RESPONSE TO REFEREE 2

The paper considers the trainability of quantum kernel methods. When given a computable kernel function, kernel methods do not suffer from trainability issues (because the optimal model can always be found). However, quantum kernels are generally not exactly computable (only approximately computable). The paper shows that the four main obstacles (high expressibility, global measurements, entanglement, and noise) plaguing the trainability of quantum neural networks manifest themselves in quantum kernel methods through the exponential concentration of kernel function values. However, I don't think this implies that the quantum kernel methods cannot learn a model close to optimal is correct (this claim is made in the paper but not proven).

Strength:

- The paper proves that under four circumstances (high expressibility, global measurements, large entanglement, and sufficient noise), the quantum kernel function exponentially concentrates on a useless kernel function (1 on the diagonal and a fixed value μ on the off-diagonal).
- This exponential concentration for quantum kernel methods under certain regimes (as identified by the authors) is problematic. The exponential concentration means that the ML model is independent of the data! An ML model that does not depend on data is useless as it would produce the same prediction for essentially all inputs.

Response: We are genuinely very grateful to the referee for this thoughtful and valuable feedback. In particular, we thank the referee for bringing to our attention these subtleties on the consequences of exponential concentration for kernel methods. We have had a lot of fun chewing it over (hence the delayed response) and now largely agree with the referee.

In essence, we agree that the main problem with exponential concentration is that in practise all estimates of kernel values (and therefore the predictions of the resulting trained model) will be dominated by finite shot effects and thus *independent* of the input training data. The training outputs are (in the limit of low regularization) typically trivially hard-coded into the model via the optimization process (independently of the kernel values), and so the model can be trivially viewed to be well trained. On the other hand, it follows from the data-independence of the kernel values that the predictions of the trained model will be completely independent of the training data and so the trained model in general performs trivially on unseen data. That is, the model generalizes terribly. However, this has a different flavor to typical barriers to generalization in that crucially it is not resolved by training on more input data points but rather by increasing the number of measurement shots. Note, the case will be more subtle when a large regularization is used but the general conclusions are the same.

Exactly how the data independence manifests depends on the kernel used and the measurement strategy taken to estimate it. For example, when we employ an overlap test to estimate the fidelity kernel, a statistical estimate of the kernel for any input data pair is found to be zero with high probability. On the other hand, for the SWAP test the kernel value is statistically indistinguishable from that obtained from random sampling. (This is sketched in our new Fig.2). We rigorously work through the different cases in our new Appendices B and C, which we summarize in Section II of the main text.

Changes: The following changes have been made in the manuscript.

1. We now formalize the key point that, with the polynomial number of measurement shots, the kernel estimates are independent of data and, consequently, the trained model also becomes insensitive to unseen input data. Our findings are summarized in Section II and proven in Appendices B and C. To add to our previous results, we concretely prove that with polynomial shots the trained model output is statistically indistinguishable from a meaningless output. We support these new analytic results with a numerical study.
2. At the end of section II, and in the discussion section, we discuss whether the problems caused by exponential concentration should be viewed as a trainability problem or generalization problem. Fig. 4 provides numerical evidence to confirm our discussion.
3. We have dropped the mention of trainability from the paper title and carefully addressed any statements concerning (un)trainability in the manuscript.

- The study on when would the exponential concentration arises is very interesting. A circumstance (global measurements) has been identified in a prior work published in Nature Comm [1]. This work identifies three more circumstances that this problem could arise.

Response: We are glad that the reviewer agrees that our systematic study of when exponential arises is interesting. We also agree that Nature Comm [1] was the first paper to highlight that global measurements can lead to exponential concentration in kernels. We now highlight this in our new Appendix A where we sit our work in the context of prior work.

Changes: We have added a new Appendix A discussing contributions from prior work.

Weakness:

- When exponential concentration happens (with off-diagonal values μ not equal to 1), the quantum kernel method could still find an approximately optimal model. When we compare the approximately computed kernel function and the exactly computed kernel function, the corresponding alpha vector will not differ much because the exact kernel matrix is positive definite (with the smallest eigenvalue equal to $1 - \mu$). The positive definiteness ensures that the optimal model does not fluctuate too much with respect to noise in the kernel entries. Hence the statement that the quantum kernel methods cannot learn a model close to optimal is incorrect.
- The statement that it is hard to distinguish between K and K_0 is correct, but that does not imply the trainability issue stated in the work. Because of the above reasoning regarding positive definiteness, I don't think quantum kernel methods have trainability issues (the inability to find a close-to-optimal model) in general (the concentrated value μ is typically close to zero). Exponential concentration is problematic (as stated by the authors), but it does not result in untrainability.

Response: As discussed above, we now agree with the referee's point that exponential concentration in the context of kernels leads to an issue that has more of a flavour of a generalization issue than trainability issue. We are grateful for the referee for bringing this to our attention.

Changes: Same as for point 1.

Overall evaluation:

- Because of the incorrect claim, I would not suggest acceptance of the work in its current form. However, the exponential concentration in quantum kernel methods is a serious problem worthy of further emphasis and study. This work provides a solid analysis of this phenomenon. I would consider acceptance if the authors could properly address the incorrect statements in the work.

Response: Thanks to the referee's very helpful comments and an opportunity to revise our work, we believe that we have now addressed these subtleties.

III. ADDITIONAL CHANGES

The following changes have been made in the revised manuscript.

1. Kernel ridge regression has been added as a new example of kernel methods at the end of Sec II A. We also edited some wordings to make the framework more rigorous.
2. The discussion about related work regarding expressivity at the end of Sec II C 1. has been moved and incorporated into Appendix A.
3. In Appendix C 3, we have added a new result showing that multiple copies of a concentrated quantum state remain indistinguishable. This provides some further intuition that error mitigation techniques can be used to avoid exponential concentration.

4. The old Appendix A has now been merged as a subsection of new Appendix C. The old analysis of model concentration has been removed as this becomes redundant thanks to the new analysis.
5. In accordance to the previous point, Lemma 1 in Appendix A of the old version has been moved to Appendix I as Supplemental Lemma 8 of the current version (where this result has been utilized).
6. In addition to new references directly related to our work in Appendix A, we have also cited some recent works including Ref. [9,10] for a quantum advantage in kernel methods as well as Ref. [76,77] for limitations of error mitigation techniques.

REVIEWER COMMENTS

Reviewer #1 (Remarks to the Author):

The authors have answered to all my comments in a satisfactorily way.

As an answer to the other reviewer's comments, the paper has been extended with a statistical analysis of the effect of the exponential concentration on the accuracy of models which are constructed by a quantum-kernel-based algorithm, in terms of generalisation property. This is in fact a more appropriate way to interpret the effects of concentration than the notion of trainability discussed in the previous version.

The argument is that in the presence of exponential concentration, clearly it is necessary to make the estimation error for a quantum expected value exponentially small in order to be able to distinguish close results. Therefore with only a polynomial number of shots the kernel estimation can be so bad as to make the model useless when used on unseen data. In theory the learnt model is optimal but, due to the shot noise, it is effectively independent of the data.

This is formally shown in two propositions for fidelity kernels in both cases of the use of overlap and swap test.

The proposition sounds reasonable to me although I was a bit puzzled by their formulation where the authors assume that the exponential concentration is toward some 'exponentially fixed small value'. What does this exactly mean? And, given that it is an essential premise for proving the propositions, should Definition 1 be modified accordingly?.

The proof of these propositions are give in an extensive appendix A, which also includes a section on related work in response to one of my comment in the first review round (wonder why this was not included in the main text).

In this appendix, I think the phrase on the first line on pag 26, 'the probability that' is no complete.

There are other typos that the authors can easily spot by reading again the new material.

In general, the paper has been improved, the new results seems reasonable although I did

not check the detailed proofs in the appendix, which I would leave to some expert in statistics.

Reviewer #2 (Remarks to the Author):

I appreciate the authors for taking my previous comments into serious considerations.

In the original submission, there was the issue that the authors describe quantum kernel methods to be untrainable, which is incorrect even when there is exponential concentration. In the revised version, the authors have addressed this incorrect claim throughout the manuscript.

However, there are a few places that the authors missed (I found these by searching the prefix "traina-", so perhaps there are more that I missed). In these places, the authors could use the word "generalization" or "predictability" in replacement of "trainability", change "is untrainable" to "will fail to generalize to unseen data", and remove statements like "exponentially flattening the training landscape".

Page 10, Below Corollary 2:

[...] highly entangling encodings should be avoided to ensure trainability.

Page 14, Second paragraph:

Here we study the trainability of TA.

Page 14, Below Proposition 4:

hence $TA(\theta)$ is untrainable with a polynomial number of measurement shots.

Page 59, Below 4. Noise:

Noise negatively affects the trainability of the parametrized quantum kernels, exponentially flattening the training landscape.

After addressing these (which I believe the authors missed unintentionally), I am happy to

suggest acceptance.

Reviewer #3 (Remarks to the Author):

I think it is important to make more explicit and formal the step where you relate the closeness of the two kernels in the Supplemental Proposition 4 (for example) to the testing of distributions in the Appendix B.

Appendix B is correct but I find the text imprecise and far too long. Sections 2 and 3 can be simple consequences of the Supplemental Lemmas 2 and 3. The test problem should be formalized using test hypothesis: the null hypothesis $H_0 : S \sim \mathcal{P}$ and the alternative hypothesis $H_1 : S \sim \mathcal{Q}$. Instead of "guess correct", we could say "right decision". The proof lines (B10) to (B15) are too detailed.

Appendix C

page 26: recall the definition of μ and σ before (C5). In line (C8)-(C10) are you sure $\mu + \sqrt{\sigma} < 1$? Otherwise, the lower bound makes no sense!

Is N the N_S ?

In (C14): is $\sigma < 1$? What is the connexion for going from (C14) to (C15)?

(C22) and (C23) are there typos: a "+" instead of a "-"?

Supplemental Proposition 4 (and other places): "exponentially fixed small value μ_0 " makes no sense! It is either fixed and arbitrarily small, or it is exponential with respect to n or N_S (!) and it is not fixed!

Can you replace "if we are randomly given" and "probability of guessing correctly" by something more precise!? This step is crucial in order to explain your reasoning as to why testing proves the impossibility of generalization. So far, it is not convincing! If you mean to do a test please describe: the test hypothesis, the observed random variables helping to make a decision. The test hypothesis cannot be random variables related to the observations that you use to make a decision.

Also, how is this test problem related to those in the Appendix B? Are you "guessing" a distribution?

Figure 11: the caption is not clear?! Is N_s fixed to 25 or varying as the graphics seem to show?!

Response to Referee Report of “Exponential concentration in quantum kernel methods”

Supanut Thanasilp,^{1,2} Samson Wang,³ M. Cerezo,^{4,5} and Zoë Holmes^{4,2}

¹*Centre for Quantum Technologies, National University of Singapore, 3 Science Drive 2 117543, Singapore.*

²*Institute of Physics, Ecole Polytechnique Fédérale de Lausanne (EPFL), CH-1015 Lausanne, Switzerland*

³*Imperial College London, London, UK.*

⁴*Information Sciences, Los Alamos National Laboratory, Los Alamos, NM, USA.*

⁵*Quantum Science Center, Oak Ridge, TN 37931, USA*

I. RESPONSE TO REFEREE 1

The authors have answered to all my comments in a satisfactorily way.

As an answer to the other reviewer’s comments, the paper has been extended with a statistical analysis of the effect of the exponential concentration on the accuracy of models which are constructed by a quantum-kernel-based algorithm, in terms of generalisation property. This is in fact a more appropriate way to interpret the effects of concentration than the notion of trainability discussed in the previous version. The argument is that in the presence of exponential concentration, clearly it is necessary to make the estimation error for a quantum expected value exponentially small in order to be able to distinguish close results. Therefore with only a polynomial number of shots the kernel estimation can be so bad as to make the model useless when used on unseen data. In theory the learnt model is optimal but, due to the shot noise, it is effectively independent of the data. This is formally shown in two propositions for fidelity kernels in both cases of the use of overlap and swap test.

In general, the paper has been improved, the new results seems reasonable although I did not check the detailed proofs in the appendix, which I would leave to some expert in statistics.

Response: We are glad that the referee is happy with our revised manuscript and appreciates the new insight on how exponential concentration affects the performance of quantum kernels. We also thanks the referee for a suggestion of an expert to check our proofs for these new results. The intuitive summary of our argument (summarised well by the reviewer above) remains unchanged in the manuscript and thanks to the third referee’s suggestion we have made a few conceptual clarifications to our proofs in the appendices. Please see the response to the third referee below for more details.

The proposition sounds reasonable to me although I was a bit puzzled by their formulation where the authors assume that the exponential concentration is toward some ‘exponentially fixed small value’. What does this exactly mean? And, given that it is an essential premise for proving the propositions, should Definition 1 be modified accordingly?.

Response: We thank the reviewer for raising this question. We agree that our previous phrasing ‘exponentially fixed small value’ was potentially confusing. What we meant was that the kernel value exponentially concentrates (over

the input data pair \mathbf{x} and \mathbf{x}') towards the concentration point μ where μ is independent of the input data pair \mathbf{x} and \mathbf{x}' and μ also vanishes exponentially with increasing qubit number (i.e., $\mu \in \mathcal{O}(b^{-n})$ for some $b > 1$).

Changes: To address this concern, we have made the following changes.

1. We have added the following clarification as a part of the modified definition 1 in the manuscript:
“[...] In addition, if μ exponentially vanishes in the number of qubits i.e., $\mu \in \mathcal{O}(1/b'^n)$ for some $b' > 1$, we say that $X(\alpha)$ exponentially concentrates towards an exponentially small value.”
2. We have replaced ‘exponentially fixed small value’ with ‘exponentially small value’ or ‘exponentially small concentration point’.
3. We have added a few additional comments in Sec. C (sources of concentration) to explicitly say that the exponential concentration in fidelity kernels also comes with the concentration point that vanishes in the number of qubits. In correspondence to this, a proof of the global-measurement-induced concentration is modified to include the computation of the concentration point.
4. We also remind readers about the definition of exponential concentration and remark on the exponentially small concentration point in Appendix C before going into the detailed discussion and proofs of practical implications of kernel concentration.

The proof of these propositions are give in an extensive appendix A, which also includes a section on related work in response to one of my comment in the first review round (wonder why this was not included in the main text).

Response/Changes: We summarize the role of prior work in the final paragraph of the introduction and then highlight that a more detailed discussion of prior work is given in the Appendix. We chose this formatting because it allowed us to have a more detailed discussion of prior work than would have been viable had we only discussed prior work in the main text.

In this appendix, I think the phrase on the first line on pag 26, ‘the probability that’ is no complete. There are other typos that the authors can easily spot by reading again the new material.

Response/Changes: We thanks the reviewer for spotting the typos. We have done further careful proof read on the new material and hopefully fixed all the remaining typos.

II. RESPONSE TO REFEREE 2

I appreciate the authors for taking my previous comments into serious considerations.

In the original submission, there was the issue that the authors describe quantum kernel methods to be untrainable, which is incorrect even when there is exponential concentration. In the revised version, the authors have addressed this incorrect claim throughout the manuscript.

After addressing these (which I believe the authors missed unintentionally), I am happy to suggest acceptance.

Response: We are happy to see that the referee’s response to our revised manuscript and suggestion for publication. As mentioned in the previous response, we genuinely appreciate the referee’s constructive comments on how to interpret the consequence of exponential concentration, which significantly improved our manuscript.

However, there are a few places that the authors missed (I found these by searching the prefix "traina-", so perhaps there are more that I missed). In these places, the authors could use the word "generalization" or "predictability" in replacement of "trainability", change "is untrainable" to "will fail to generalize to unseen data", and remove statements like "exponentially flattening the training landscape".

Page 10, Below Corollary 2: [...] highly entangling encodings should be avoided to ensure trainability.

Page 14, Second paragraph: Here we study the trainability of TA.

Page 14, Below Proposition 4: hence $\text{TA}(\theta)$ is untrainable with a polynomial number of measurement shots.

Page 59, Below 4. Noise: Noise negatively affects the trainability of the parametrized quantum kernels, exponentially flattening the training landscape.

Response/Changes: We thanks the referee for pointing this out. Indeed, there are few more places where we have missed the correction. We have gone through again and hopefully sorted out the rest.

We would like to clarify that the trainability language used on Page 14 and 59 mentioned above is intentional as it describes the training of the parameterized embedding. In this case, similarly to training parameterized quantum circuits, one actually needs to optimize a non-convex landscape. We have clarified this potential confusion in the manuscript.

III. RESPONSE TO REFEREE 3

I think it is important to make more explicit and formal the step where you relate the closeness of the two kernels in the Supplemental Proposition 4 (for example) to the testing of distributions in the Appendix B.

Response: We are very appreciative of the referee's constructive feedback on the presentation of our proofs. We have made our reasoning more explicit and revised the formal statements of our statements and proofs. Please see below for more details.

Appendix B is correct but I find the text imprecise and far too long. Sections 2 and 3 can be simple consequences of the Supplemental Lemmas 2 and 3. The test problem should be formalized using test hypothesis: the null hypothesis $H_0 : S \sim P$ and the alternative hypothesis $H_1 : S \sim Q$. Instead of "guess correct", we could say "right decision". The proof lines (B10) to (B15) are too detailed.

Response: We thanks the referee for the helpful comments. We have now improved the presentation of Appendix B to be more formal and concise. We appreciate that it may still be a little long to the reviewers taste; however, we would like to stress that many researchers in the quantum machine learning community will not be so familiar with hypothesis testing and we would like our work to be accessible.

Changes:

1. Supplemental Lemma 1. and Supplemental Proposition 1. are now written in terms of the test hypotheses, with the null and alternative hypotheses being specified.
2. As suggested, we have changed the wording from "guess correct" to "make right decision" in Appendix B and also in Appendix C.
3. We have made Supplemental Lemma 2. more specific to our scenario later (by being more specific about the product distributions $\mathcal{P}^{\otimes N}$ and $\mathcal{Q}^{\otimes N}$, instead of $\mathcal{P} = \prod_{i=1}^N \mathcal{P}_i$ and $\mathcal{Q} = \prod_{i=1}^N \mathcal{Q}_i$). In addition, we have shortened the proof lines (B10) to (B15) in the previous version, as suggested by the referee.
4. We have removed Section 3 of Appendix B which now becomes redundant in the new presentation style.

Appendix C

page 26: recall the definition of μ and σ before (C5). In line (C8)-(C10) are you sure $\mu + \sqrt{\sigma} < 1$? Otherwise, the lower bound makes no sense!

In (C14): is $\sigma < 1$? What is the connexion for going from (C14) to (C15)?

Response/Changes: We thanks for the referee’s suggestion and question. To improve the readability of the proof, we have made some modifications. We now first state explicitly that we assume kernel concentration and the corresponding scaling of μ and β . We note that to keep the consistency in the notation with the definition of exponential concentration (i.e., Definition 1) we replace σ^2 in the previous version with β (i.e., $\beta = \sigma^2$).

With this new structure of the proof, it also directly clarifies the previous confusion from Eq. (C14) to Eq. (C15). To be more precise, in the old version, (C15) is a continuation from (C10), and (C14) is an explanation from (C8) to (C9). This has been addressed in the new version where (C14) (in the old version) has been moved to the beginning of the proof in the revised version.

We can confirm that $\mu + \sqrt{\sigma} < 1$ and $\sigma < 1$ as in this section we are investigating the effects of exponential concentration and so we assume here that the kernel values exponentially concentrate to the exponentially small concentration point (Definition 1.). That is, we have $\mu \in \mathcal{O}(1/b'^n)$ for some $b' > 1$ and $\sigma^2 \in \mathcal{O}(1/b^n)$ for some $b > 1$ (where n is the number of qubits). Both vanish exponentially and become exponentially smaller than 1 at large n .

Is N the N_s ?

Response/Changes: We denote N for the number of samples/measurement outcomes and N_s for the number of training data points. We can see how this would easily confuse readers. To address this concern, we have added reminders in different places for what N and N_s stand for, particularly in our formal statements (e.g., Supplemental Proposition 2) as well as in figure’s captions/axes.

(C22) and (C23) are there typos: a ”+” instead of a ”-”?

Response/Changes: We thank the referee for spotting this. Indeed, it is a typo.

Supplemental Proposition 4 (and other places): ”exponentially fixed small value μ_0 ” makes no sense! It is either fixed and arbitrarily small, or it is exponential with respect to n or N_s (!) and it is not fixed!

Response: We are grateful for this helpful feedback- indeed the phrasing “exponentially fixed small value” is inappropriate. What we intended to say here is that μ is a concentration point (that is independent of the input data pair) and it vanishes exponentially with the number of qubits. Please also see our response to the same concern by the first referee.

Response: We have made the following changes regarding this concern. Note these changes are also mentioned in the response to the first referee and repeated here for the referee’s convenience.

1. We have added the following clarification as a part of the modified definition 1 in the manuscript:
“[...] In addition, if μ exponentially vanishes in the number of qubits i.e., $\mu \in \mathcal{O}(1/b'^n)$ for some $b' > 1$, we say that $X(\alpha)$ exponentially concentrates towards an exponentially small value.”
2. We have replaced ‘exponentially fixed small value’ with ‘exponentially small value’ or ‘exponentially small concentration point’.

3. We have added a few additional comments in Sec. C (sources of concentration) to explicitly say that the exponential concentration in fidelity kernels also comes with the concentration point that vanishes in the number of qubits. In correspondence to this, a proof of the global-measurement-induced concentration is modified to include the computation of the concentration point.
4. We also remind readers about the definition of exponential concentration and remark on the exponentially small concentration point in Appendix C before going into the detailed discussion and proofs of practical implications of kernel concentration.

Can you replace “if we are randomly given” and “probability of guessing correctly” by something more precise!? This step is crucial in order to explain your reasoning as to why testing proves the impossibility of generalization. So far, it is not convincing! If you mean to do a test please describe: the test hypothesis, the observed random variables helping to make a decision. The test hypothesis cannot be random variables related to the observations that you use to make a decision.

Also, how is this test problem related to those in the Appendix B? Are you “guessing” a distribution?

Response/Changes: We are grateful to the referee for this constructive feedback and the opportunity for us to improve the rigor of our results. We have largely revised Appendix C2. by being more explicit and precise in our reasoning, formal statements and our proofs. In particular, to address the questions raised by the referee, we have made the following specific changes.

1. We have replaced “if we are randomly given” with “a set of samples drawn from either ... or ... (with an equal probability) ”, and “probability of guessing correctly” with “probability of making right decision”.
2. We formally introduce the statistical indistinguishability between two distributions \mathcal{P} and \mathcal{Q} (which was left implicit in the previous version) i.e., Definition 2. This is defined in terms of a hypothesis test with N samples (drawn from either \mathcal{P} or \mathcal{Q} with equal probability) where the null and alternative hypotheses are clearly defined. With this formal definition, the connection to Appendix B is hopefully clarified.
3. We also define a notation of the statistical indistinguishability of the outcomes (i.e., a post processing of samples) in Definition 3. This introduces the language for us to talk about the indistinguishability of the estimated kernel values/model predictions more formally.

Figure 11: the caption is not clear?! Is N_s fixed to 25 or varying as the graphics seem to show?!

Response/Changes: We thanks the referee for spotting this. Indeed, this is a typo from our side and we have now fixed this. It is supposed to be N (the number of samples) instead of N_s (the number of training data). We vary the number of samples and fix the training data.

IV. ADDITIONAL CHANGES

The following changes have been made in the revised manuscript.

1. Since Appendix C becomes relatively long, we have added a summary of the section at the beginning.
2. To improve readability, we have re-structured Appendix C2 by grouping all the proofs of the formal statements to the end of the subsection.
3. In accordance with the changes made in the SWAP test for the fidelity kernel, we have propagated this change to the projected kernel section in Appendix C2.